# Metabolic co-dependence drives the evolutionarily ancient *Hydra–Chlorella* symbiosis

Mayuko Hamada[1,2†], Katja Schröder[3,4†], Jay Bathia[3,4], Ulrich Kürn[3,4], Sebastian Fraune[3,4], Mariia Khalturina[1], Konstantin Khalturin[1], Chuya Shinzato[1,5], Nori Satoh[1], Thomas CG Bosch[3,4*]

[1]Marine Genomics Unit, Okinawa Institute of Science and Technology Graduate University, Okinawa, Japan; [2]Ushimado Marine Institute, Okayama University, Okayama, Japan; [3]Interdisciplinary Research Center, Kiel Life Science, Kiel University, Kiel, Germany; [4]Zoological Institute, Kiel Life Science, Kiel University, Kiel, Germany; [5]Atmosphere and Ocean Research Institute, The University of Tokyo, Tokyo, Japan

**Abstract** Many multicellular organisms rely on symbiotic associations for support of metabolic activity, protection, or energy. Understanding the mechanisms involved in controlling such interactions remains a major challenge. In an unbiased approach we identified key players that control the symbiosis between *Hydra viridissima* and its photosynthetic symbiont *Chlorella* sp. A99. We discovered significant up-regulation of *Hydra* genes encoding a phosphate transporter and glutamine synthetase suggesting regulated nutrition supply between host and symbionts. Interestingly, supplementing the medium with glutamine temporarily supports in vitro growth of the otherwise obligate symbiotic *Chlorella*, indicating loss of autonomy and dependence on the host. Genome sequencing of *Chlorella* sp. A99 revealed a large number of amino acid transporters and a degenerated nitrate assimilation pathway, presumably as consequence of the adaptation to the host environment. Our observations portray ancient symbiotic interactions as a codependent partnership in which exchange of nutrients appears to be the primary driving force.
DOI: https://doi.org/10.7554/eLife.35122.001

**\*For correspondence:**
tbosch@zoologie.uni-kiel.de

[†]These authors contributed equally to this work

**Competing interests:** The authors declare that no competing interests exist.

## Introduction

Symbiosis has been a prevailing force throughout the evolution of life, driving the diversification of organisms and facilitating rapid adaptation of species to divergent new niches (*Moran, 2007*; *Joy, 2013*; *McFall-Ngai et al., 2013*). In particular, symbiosis with photosynthetic symbionts is observed in many species of cnidarians such as corals, jellyfish, sea anemones and hydra, contributing to the ecological success of these sessile or planktonic animals (*Douglas, 1994*; *Davy et al., 2012*). Among the many animals dependent on algal symbionts, inter-species interactions between green hydra *Hydra viridissima* and endosymbiotic unicellular green algae of the genus *Chlorella* have been a subject of interest for decades (*Muscatine and Lenhoff, 1963*; *Roffman and Lenhoff, 1969*). Such studies not only provide insights into the basic 'tool kit' necessary to establish symbiotic interactions, but are also of relevance in understanding the resulting evolutionary selective processes (*Muscatine and Lenhoff, 1965a*; *1965b*; *Thorington and Margulis, 1981*).

The symbionts are enclosed in the host endodermal epithelial cells within perialgal vacuoles called 'symbiosomes'. The interactions at play here are clearly metabolic: the algae depend on nutrients that are derived from the host or from the environment surrounding the host, while in return the host receives a significant amount of photosynthetically fixed carbon from the algae.

**eLife digest** All animals host microorganisms; some of which form 'symbiotic' relationships with their host that are mutually beneficial. For instance, the human gut shelters tens of thousands of species of bacteria that break down our food for us, and corals, jellyfish or sea anemones can extract energy directly from sunlight thanks to the algae that live inside their cells.

*Hydra*, a small freshwater animal, lives in a symbiotic relationship with algae called *Chlorella* that it carries inside its cells. Once an independent organism, *Chlorella* has evolved in such a way that, in nature, it cannot exist without *Hydra* anymore. In turn, the algae produce sugars to fuel the animal when it cannot get food from the environment. Yet, despite over 30 years of research, it still remains unclear how exactly the relationship between *Hydra* and *Chlorella* works, and how it came to be. Understanding how these two organisms live together could help researchers to figure out the general principles that guide symbiotic interactions.

Nitrogen is an element that is essential for life, and organisms can extract it from various sources, such as nitrates or the amino acid glutamine. Here, Hamada, Schröder et al. sequenced the entire genome of *Chlorella*. This revealed that *Chlorella* has lost someof the genes required to obtain nitrates, and to process them into nitrogen. However, the genetic analysis showed that the algae express genes that allow them to import amino acids.

In turn, analysis of the genes expressed by *Hydra* when it lives in symbiosis with *Chlorella* showed that the animal turns on genetic information needed to make glutamine. It thus seems that *Hydra* creates glutamine which *Chlorella* can import; the algae then process this amino acid to obtain the nitrogen they need. Hamada, Schröder et al. also discovered that if the environment was artificially enriched in glutamine, *Chlorella* could live on their own outside of *Hydra* for a while.

The results suggest that symbiotic relationships, such as the one between *Hydra* and *Chlorella*, were established because the organisms became dependent on each other for essential nutrients. This co-dependency is strengthened if the organisms lose the ability to produce the nutrients on their own. However, this partnership may be altered when the environment changes too much, especially if the balance of nutrients available gets tipped. For example, if seas that are normally poor in nutrients become suddenly rich in these elements, this may disrupt the existence of symbiotic organisms such as corals.

DOI: https://doi.org/10.7554/eLife.35122.002

Previous studies have provided evidence that the photosynthetic symbionts provide their host with maltose, enabling *H. viridissima* to survive periods of starvation (*Muscatine and Lenhoff, 1963*; *Muscatine, 1965*; *Roffman and Lenhoff, 1969*; *Cook and Kelty, 1982*; *Huss et al., 1994*). *Chlorella*-to-*Hydra* translocation of photosynthates is critical for polyps to grow (*Muscatine and Lenhoff, 1965b*; *Mews, 1980*; *Douglas and Smith, 1983*; *1984*). Presence of symbiotic algae also has a profound impact on hydra´s fitness by promoting oogenesis (*Habetha et al., 2003*; *Habetha and Bosch, 2005*).

Pioneering studies performed in the 1980 s (*McAuley and Smith, 1982*; *Rahat and Reich, 1984*) showed that there is a great deal of adaptation and specificity in this symbiotic relationship. All endosymbiotic algae found in a single host polyp are clonal and proliferation of symbiont and host is tightly correlated (*Bossert and Dunn, 1986*; *McAuley, 1986a*). Although it is not yet known how *Hydra* controls cell division in symbiotic *Chlorella*, *Chlorella* strain A99 is unable to grow outside its polyp host and is transmitted vertically to the next generation of *Hydra*, indicating loss of autonomy during establishment of its symbiotic relationship with this host (*Muscatine and McAuley, 1982*; *Campbell, 1990*; *Habetha et al., 2003*).

Molecular phylogenetic analyses suggest that *H. viridissima* is the most basal species in the genus *Hydra* and that symbiosis with *Chlorella* was established in the ancestral *viridissima* group after their divergence from non-symbiotic *Hydra* groups (*Martínez et al., 2010*; *Schwentner and Bosch, 2015*). A recent phylogenetic analysis of different strains of green hydra resulted in a phylogenetic tree that is topologically equivalent to that of their symbiotic algae (*Kawaida et al., 2013*), suggesting these species co-evolved as a result of their symbiotic relationship. Although our understanding of the factors that promote symbiotic relationships in cnidarians has increased (*Shinzato et al.,*

*2011*; *Davy et al., 2012*; *Lehnert et al., 2014*; *Baumgarten et al., 2015*; *Ishikawa et al., 2016*), very little is known about the molecular mechanisms allowing this partnership to persist over millions of years.

Recent advances in transcriptome and genome analysis allowed us to identify the metabolic interactions and genomic evolution involved in achieving the *Hydra-Chlorella* symbiotic relationship. We present here the first characterization, to our knowledge, of genetic complementarity between green *Hydra* and *Chlorella* algae that explains the emergence and/or maintenance of a stable symbiosis. We also provide here the first report of the complete genome sequence from an obligate intracellular *Chlorella* symbiont. Together, our results show that exchange of nutrients is the primary driving force for the symbiosis between *Chlorella* and *Hydra*. Subsequently, reduction of metabolic pathways may have further strengthened their codependency. Our findings provide a framework for understanding the evolution of a highly codependent symbiotic partnership in an early emerging metazoan.

## Results

### Discovery of symbiosis-dependent *Hydra* genes

As tool for our study we used the green hydra *H. viridissima* (*Figure 1A*) colonized with symbiotic *Chlorella* sp. strain A99 (abbreviated here as Hv_Sym), aposymbiotic *H. viridissima* from which the symbiotic *Chlorella* were removed (Hv_Apo), as well as aposymbiotic *H.viridissima*, which have been artificially infected with *Chlorella variabilis* NC64A (Hv_NC64A). The latter is symbiotic to the single-cellular protist *Paramecium* (*Karakashian and Karakashian, 1965*). Although an association between *H. viridissima* and *Chlorella* NC64A can be maintained for some time, both their growth rate (*Figure 1B*) and the number of NC64A algae per *Hydra* cell (*Figure 1—figure supplement 1*) are significantly reduced compared to the symbiosis with native symbiotic *Chlorella* A99.

H.*H. viridissima* genes involved in the symbiosis with *Chlorella* algae were identified by microarray based on the contigs of *H. viridissima* A99 transcriptome (NCBI GEO Platform ID: GPL23280). For the microarray analysis, total RNA was extracted from the polyps after light exposure for six hours. By comparing the transcriptomes of Hv_Sym and Hv_Apo, we identified 423 contigs that are up-regulated and 256 contigs that are down-regulated in presence of *Chlorella* A99 (*Figure 1C*). To exclude genes involved in oogenesis and embryogenesis, only contigs differently expressed with similar patterns in both sexual and asexual Hv_Sym were recorded. Interestingly, contigs whose predicted products had no discernible homologs in other organisms including other *Hydra* species were overrepresented in these differentially expressed contigs (Chi-squared test p<0.001) (*Figure 1—figure supplement 2*). Such taxonomically restricted genes (TRGs) are thought to play important roles in the development of evolutionary novelties and morphological diversity within a given taxonomic group (*Khalturin et al., 2009*; *Tautz and Domazet-Lošo, 2011*).

We further characterized functions of the differentially expressed *Hydra* genes by Gene Ontology (GO) terms (*Ashburner et al., 2000*) and found the GO term 'localization' overrepresented among up-regulated contigs (Hv_Sym > Hv_Apo), whereas the GO term 'metabolic process' was enriched among down-regulated contigs (Hv_Sym < Hv_Apo) (*Figure 1D*). More specifically, the up-regulated contigs included many genes related to 'transmembrane transporter activity', 'transmembrane transport', 'transposition', 'cilium' and 'protein binding, bridging' (*Figure 1E*). In the down-regulated contig set, the GO classes 'cellular amino acid metabolic process', 'cell wall organization or biogenesis' and 'peptidase activity' were overrepresented (*Figure 1E*). These results suggest that the *Chlorella* symbiont affects core metabolic processes and pathways in *Hydra*. Particularly, carrier proteins and active membrane transport appear to play a prominent role in the symbiosis.

As next step, we used GO terms, domain search and similarity search to further analyze the differentially expressed contigs between Hv_Sym and Hv_Apo (*Supplementary file 1*). As the genes with GO terms related to localization and transport, we identified 27 up-regulated contigs in Hv_Sym (*Table 1*). Interestingly, this gene set included a contig showing sequence similarity to the glucose transporter GLUT8 gene, which was previously reported to be up-regulated in the symbiotic state of the sea anemone *Aiptasia* (*Lehnert et al., 2014*; *Sproles et al., 2018*). Thus, a conserved mechanism may be responsible for photosynthate transport from the symbiont into the host cytoplasm across the symbiosome membrane. Further, a contig encoding a carbonic anhydrase (CA) enzyme was up-

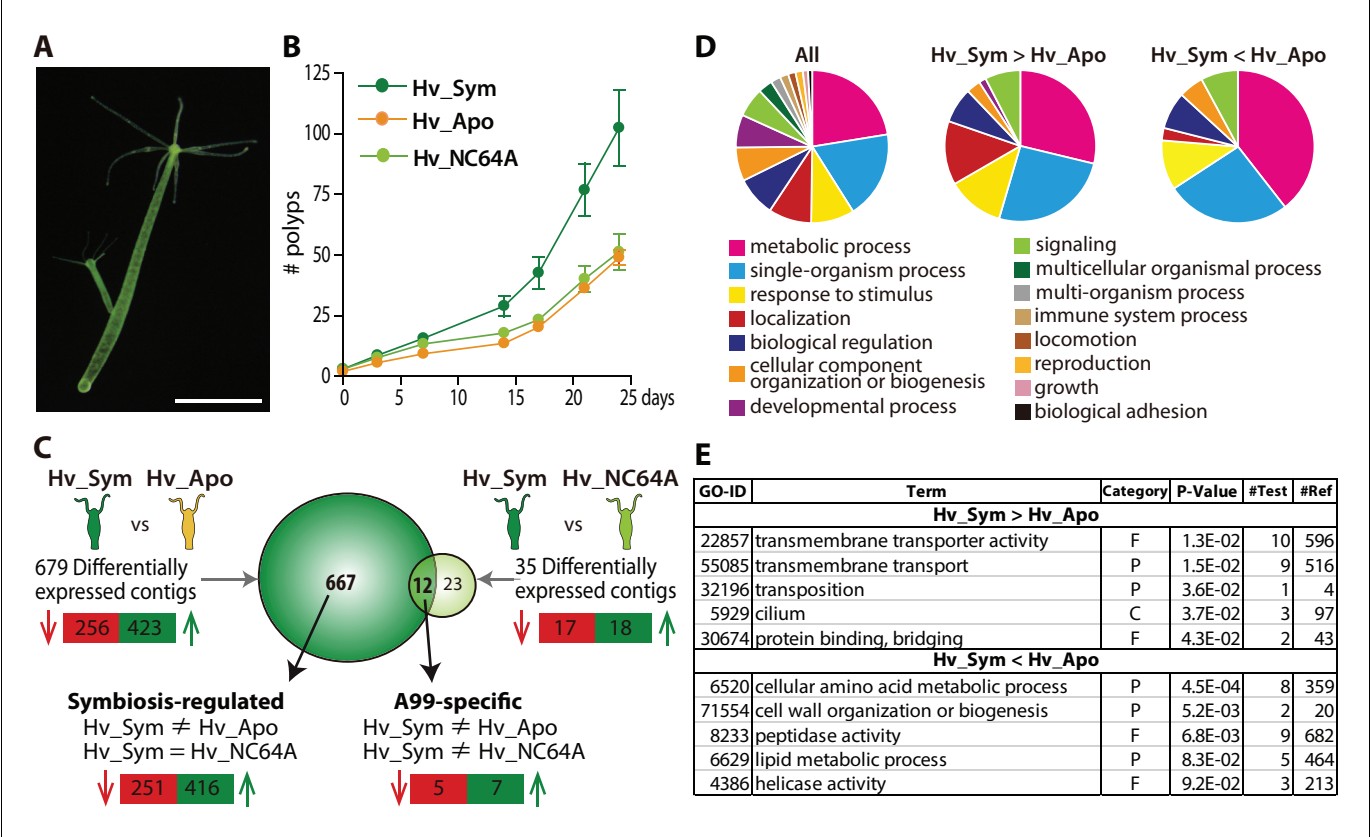

**Figure 1.** *Hydra* growth and differential expression of *Hydra* genes resulting from symbiosis. (**A**) *Hydra viridissima* strain A99 used for this study. Scale bar, 2 mm. (**B**) Growth rates of polyps grown with native symbiotic *Chlorella* A99 (Hv_Sym, dark green), Aposymbiotic polyps from which *Chlorella* were removed (Hv_Apo, orange) and aposymbiotic polyps reinfected with *Chlorella variabilis* NC64A (Hv_NC64A, light green). Average of the number of hydra in each experimental group (n = 6) is represented. Error bars indicate standard deviation. (**C**) Graphic representation of differentially expressed genes identified by microarray. The transcriptome of Hv_Sym is compared with that of Hv_Apo and Hv_NC64A with the number of down-regulated contigs in Hv_Sym shown in red and those up-regulated in green. Genes differentially expressed in Hv_Sym compared to both Hv_Apo and Hv_NC64A are given as 'A99-specific', those differentially expressed between Hv_A99 and Hv_Apo but not Hv_NC64A as 'Symbiosis-regulated'. (**D**) GO distribution of Biological Process at level two in all contigs (All), up-regulated contigs (Hv_Sym > Hv_Apo) and down-regulated contigs (Hv_Sym < Hv_Apo) in Hv_Sym. (**E**) Overrepresented GO terms in up-regulated contigs (Hv_Sym > Hv_Apo) and down-regulated contigs (Hv_Sym < Hv_Apo). Category, F: molecular function, C: cellular component, P: biological process. P-values, probability of Fisher's exact test. #Test, number of corresponding contigs in differentially expressed contigs. #Ref, number of corresponding contigs in all contigs.

DOI: https://doi.org/10.7554/eLife.35122.003

The following source data and figure supplements are available for figure 1:

**Source data 1.** GO distribution of Biological Process in all contigs (All), up-regulated contigs (up: Hv_Sym > Hv_Apo) and down-regulated contigs (down: Hv_Sym < Hv_Apo) in Hv_Sym.
DOI: https://doi.org/10.7554/eLife.35122.007

**Figure supplement 1.** *Chlorella* sp. A99 and *Chlorella variabilis* NC64A in *Hydra viridissima* A99.
DOI: https://doi.org/10.7554/eLife.35122.004

**Figure supplement 2.** Conserved genes and species-specific genes differentially expressed in symbiotic *Hydra*.
DOI: https://doi.org/10.7554/eLife.35122.005

**Figure supplement 3.** Glutamine synthetase (GS) genes in Cnidarians.
DOI: https://doi.org/10.7554/eLife.35122.006

regulated in Hv_Sym (*Table 1*). CA catalyzes the interconversion of $HCO_3$ and $CO_2$. Similar to the GLUT8 gene, carbonic anhydrase also appears to be up-regulated in symbiotic corals and anemones (*Weis et al., 1989*; *Grasso et al., 2008*; *Ganot et al., 2011*; *Lehnert et al., 2014*). It appears plausible that for efficient photosynthesis in symbiotic algae, the host may need to convert $CO_2$ to the less freely diffusing inorganic carbon ($HCO_3$) to maintain it in the symbiosome (*Lucas and Berry, 1985*; *Weis et al., 1989*; *Barott et al., 2015*). We also observed up-regulation of contigs encoding

**Table 1.** List of differentially expressed genes between Hv_Sym and Hv_Apo, which are likely to be involved in symbiotic relationship

| Probename | Fold change | | | Human_BestHit | blast2GO_Description |
|---|---|---|---|---|---|
| | Hv_Sym /Hv_Apo | Hv_Sym_sexy /Hv_Apo | Hv_NC64A /Hv_Sym | | |
| Localization and Transport | | | | | |
| Hv_Sym > Hv_Apo | | | | | |
| rc_6788 | 9.87 | 8.00 | 1.01 | | helicase conserved c-terminal domain containing protein |
| rc_10246 | 8.26 | 5.15 | 1.82 | | protein |
| rc_6298 | 7.10 | 4.73 | 0.99 | hypothetical protein LOC220081 | protein fam194b |
| 2268 | 6.96 | 3.58 | 1.26 | protein Daple | viral a-type inclusion protein |
| 10548 | 6.74 | 6.89 | 0.73 | transient receptor potential cation channel subfamily M member three isoform d | transient receptor potential cation channel subfamily m member 3-like |
| rc_1290 | 6.44 | 7.18 | 0.99 | tetratricopeptide repeat protein eight isoform B | tetratricopeptide repeat protein 8 |
| 18736 | 6.04 | 6.34 | 1.03 | BTB/POZ domain-containing protein KCTD9 | btb poz domain-containing protein kctd9-like; unnamed protein product |
| rc_9270 | 5.96 | 10.03 | 1.37 | PREDICTED: hypothetical protein LOC100131693 | eukaryotic translation initiation factor 4e |
| NPNHRC_15697 | 3.85 | 2.74 | 0.62 | | major facilitator superfamily domain-containing protein 1 |
| 290 | 3.68 | 3.73 | 1.32 | splicing factor, arginine/ serine-rich 6 | splicing arginine serine-rich 4 |
| rc_9596 | 3.56 | 4.19 | 1.62 | BTB/POZ domain-containing protein KCTD10 | btb poz domain-containing adapter for cul3-mediated degradation protein 3 |
| rc_6774 | 3.34 | 3.32 | 1.31 | solute carrier family 43, member 2 | large neutral amino acids transporter small subunit 4 |
| rc_26218 | 3.29 | 2.91 | 0.41 | sodium-dependent phosphate transport protein 2A isoform 1 | sodium-dependent phosphate transport protein 2b |
| NPNHRC_26094 | 3.20 | 3.98 | 1.31 | SPE-39 proteinid="T5" | spe-39 protein |
| 9096 | 3.10 | 2.20 | 0.69 | otoferlin isoform d | otoferlin |
| rc_21349 | 2.89 | 4.25 | 0.78 | 5′-AMP-activated protein kinase catalytic subunit alpha-2 | 5 -amp-activated protein kinase catalytic subunit alpha-2 |
| npRC_14488 | 2.88 | 2.65 | 0.71 | solute carrier family 2, facilitated glucose transporter member 8 | solute carrier family facilitated glucose transporter member 8-like |
| 8863 | 2.75 | 2.70 | 0.81 | ATP-binding cassette, sub-family B, member 10 precursor | abc transporter b family protein |
| rc_11896 | 2.49 | 2.56 | 1.52 | ATP-binding cassette, sub-family B, member 10 precursor | abc transporter b family member 25-like |
| rc_6842 | 2.41 | 3.35 | 1.59 | hypothetical protein LOC112752 isoform 2 | intraflagellar transport protein 43 homolog |
| 5242 | 2.36 | 3.35 | 1.22 | growth arrest-specific protein 8 | growth arrest-specific protein 8 |
| 5815 | 2.23 | 2.47 | 0.78 | plasma membrane calcium-transporting ATPase 4 isoform 4a | plasma membrane calcium atpase |
| 8765 | 2.22 | 3.25 | 0.91 | growth arrest-specific protein 8 | growth arrest-specific protein 8 |
| NPNH_14052 | 2.19 | 2.17 | 0.79 | V-type proton ATPase 21 kDa proteolipid subunit isoform 2 | v-type proton atpase 21 kda proteolipid subunit-like |
| rc_2499 | 2.18 | 2.03 | 1.47 | endoplasmic reticulum-Golgi intermediate compartment protein three isoform a | endoplasmic reticulum-golgi intermediate compartment protein 3 isoform 2 |
| rc_13969 | 2.08 | 3.09 | 0.97 | | major facilitator superfamily |
| (IPR023561) Carbonic anhydrase, alpha-class | | | | | |

*Table 1 continued on next page*

*Table 1 continued*

| Probename | Fold change | | | Human_BestHit | blast2GO_Description |
|---|---|---|---|---|---|
| | Hv_Sym /Hv_Apo | Hv_Sym_sexy /Hv_Apo | Hv_NC64A /Hv_Sym | | |
| rc_24825 | 2.49 | 2.38 | 0.83 | protein tyrosine phosphatase, receptor type, G precursor | receptor-type tyrosine-protein phosphatase gamma |
| Cell Adhesion and extracelluar matrix | | | | | |
| Hv_Sym > Hv_Apo | | | | | |
| 7915 | 4.01 | 5.09 | 0.94 | fibrillin-2 precursor | fibrillin-1- partial |
| npRC_24163 | glutamate3.69 | 3.59 | 1.32 | semaphorin 5A precursor | rhamnospondin 1 |
| Immunity, apoptosis and recognition | | | | | |
| Hv_Sym > Hv_Apo | | | | | |
| (IPR000157) Toll/interleukin-1 receptor homology (TIR) domain | | | | | |
| 5168 | 9.28 | 4.92 | 0.61 | | protein; PREDICTED: uncharacterized protein LOC100893943 |
| 12749 | 5.13 | 3.35 | 1.26 | | PREDICTED: uncharacterized protein LOC100893943 [Strongylocentrotus purpuratus] |
| (IPR011029) DEATH-like | | | | | |
| 6508 | 6.70 | 5.10 | 0.64 | | PREDICTED: hypothetical protein [Hydra magnipapillata] |
| rc_2417 | 5.39 | 2.70 | 1.01 | | nod3 partial; PREDICTED: uncharacterized protein LOC100206003 |
| (IPR002398) Peptidase C14, caspase precursor p45 | | | | | |
| NPNH_21275 | 2.36 | 3.53 | 1.18 | caspase seven isoform alpha precursor | caspase d |
| (IPR016187) C-type lectin fold | | | | | |
| 11411 | 2.93 | 2.98 | 0.75 | C-type mannose receptor 2 | PREDICTED: similar to predicted protein, partial [Hydra magnipapillata] |
| Hv_Sym < Hv_Apo | | | | | |
| (IPR000488) Death | | | | | |
| 7319 | 0.45 | 0.31 | 1.10 | probable ubiquitin carboxyl-terminal hydrolase CYLD isoform 2 | ubiquitin carboxyl-terminal hydrolase cyld |
| (IPR001875) Death effector domain | | | | | |
| RC_FV81RT001CSTY | 0.31 | 0.39 | 0.93 | astrocytic phosphoprotein PEA-15 | fadd |
| Chitinase | | | | | |
| Hv_Sym > Hv_Apo | | | | | |
| (IPR001223) Glycoside hydrolase, family 18, catalytic domain | | | | | |
| rc_4450 | 2.78 | 3.83 | 0.66 | | chitinase 2 |
| Hv_Sym < Hv_Apo | | | | | |
| (IPR000726) Glycoside hydrolase, family 19, catalytic | | | | | |
| FPVQZVL01EAWBY | 0.21 | 0.16 | 1.78 | | endochitinase 1-like |
| 1028 | 0.23 | 0.18 | 1.47 | | endochitinase 1-like |
| Oxidative Stress Response | | | | | |
| Hv_Sym > Hv_Apo | | | | | |
| np_1276 | 5.99 | 7.16 | 0.78 | glutaredoxin-2, mitochondrial isoform 2 | cpyc type |
| 10926 | 3.9 | 2.3 | 0.8 | hydroxysteroid dehydrogenase-like protein 2 | hydroxysteroid dehydrogenase-like protein 2 |

*Table 1 continued on next page*

*Table 1 continued*

| Probename | Fold change | | | Human_BestHit | blast2GO_Description |
|---|---|---|---|---|---|
| | Hv_Sym /Hv_Apo | Hv_Sym_sexy /Hv_Apo | Hv_NC64A /Hv_Sym | | |
| 469 | 2.97 | 3.53 | 0.76 | cytochrome P450 3A7 | cytochrome p450 |
| FV81RT001DCTAQ | 2.69 | 2.50 | 0.75 | oxidoreductase NAD-binding domain-containing protein one precursor | oxidoreductase nad-binding domain-containing protein 1 |
| 696 | 2.30 | 3.24 | 0.69 | methionine-R-sulfoxide reductase B1 | selenoprotein 1; methionine-r-sulfoxide reductase b1-a-like |
| 6572 | 2.23 | 2.15 | 1.06 | L-xylulose reductase | l-xylulose reductase |
| 13298 | 2.10 | 3.49 | 0.64 | eosinophil peroxidase preproprotein | peroxidase |
| npRC_6975 | 2.04 | 2.77 | 1.42 | methionine-R-sulfoxide reductase B1 | selenoprotein 1; methionine-r-sulfoxide reductase b1-a-like |
| (IPR024079) Metallopeptidase, catalytic domain | | | | | |
| Hv_array_4952 | 4.77 | 13.31 | 0.72 | meprin A subunit beta precursor | zinc metalloproteinase nas-4-like |
| Hv_array_rc_3992 | 2.66 | 2.23 | 1.27 | matrix metalloproteinase seven preproprotein | matrix metalloproteinase-24-like |
| Hv_Sym < Hv_Apo | | | | | |
| RC_FWZAEML02HKSC | 0.255 | 0.153 | 1.444 | | ascorbate peroxidase |
| np_14962 | 0.293 | 0.455 | 1.390 | tryptophan 5-hydroxylase 2 | phenylalanine hydroxylase |
| rc_4151 | 0.318 | 0.463 | 1.693 | phenylalanine-4-hydroxylase | phenylalanine hydroxylase |
| 2835 | 0.384 | 0.344 | 1.787 | | u1 small nuclear ribonucleoprotein 70 kda |
| rc_11426 | 0.413 | 0.458 | 1.591 | short-chain dehydrogenase/ reductase family 9C member 7 | uncharacterized oxidoreductase -like |
| FWZAEML02IC34R | 0.427 | 0.448 | 1.159 | aldehyde dehydrogenase 5A1 isoform two precursor | succinate-semialdehyde mitochondrial-like |
| FWZAEML02HKSCO | 0.454 | 0.307 | 0.833 | | ascorbate peroxidase |
| (IPR004045) Glutathione S-transferase, N-terminal | | | | | |
| RC_FWZAEML02GGHN | 0.09 | 0.07 | 1.81 | hematopoietic prostaglandin D synthase | glutathione s-transferase family member (gst-7) |
| (IPR024079) Metallopeptidase, catalytic domain | | | | | |
| rc_11270 | 0.14 | 0.20 | 1.33 | meprin A subunit beta precursor | protein; zinc metalloproteinase nas-4-like |
| rc_RSASM_15059 | 0.22 | 0.29 | 1.42 | | —NA— |
| 2111 | 0.37 | 0.43 | 1.74 | meprin A subunit beta precursor | zinc metalloproteinase nas-4-like |
| 12451 | 0.50 | 0.39 | 0.78 | meprin A subunit alpha precursor | zinc metalloproteinase nas-13- partial |
| (IPR013122) Polycystin cation channel, PKD1/PKD2 | | | | | |
| 28854 | 0.37 | 0.28 | 0.94 | polycystin-2 | receptor for egg jelly partial |
| 15774 | 0.40 | 0.26 | 0.76 | polycystic kidney disease protein 1-like two isoform a | protein |

DOI: https://doi.org/10.7554/eLife.35122.008

proteins involved in vesicular and endosomal trafficking, such as spe-39 protein, otoferlin, protein fam194b and V-type proton ATPase 21 kda proteolipid, which may have a function in nutrition exchange between host and symbiont and maintenance of proper condition in the symbiosome. Upregulated genes also include genes encoding rhamnospondin and fibrillin, known to be involved in cell adhesion and extracellular matrix, and retention of the symbiont at the proper site in the *Hydra* cells.

Photosynthesis by symbiotic algae imposes Reactive Oxygen Species (ROS) that can damage lipids, proteins and DNA in the host cells (*Lesser, 2006*). Therefore, in symbiosis with photosynthetic organisms an appropriate oxidative stress response of the host is required for tolerance of the symbiont. Indeed, an increase of antioxidant activities in symbiotic states of cnidarians has been reported previously (*Richier et al., 2005*) and it has been suggested that ROS produced by stress could be the major trigger of symbiosis breakdown during coral bleaching (*Lesser, 2006*; *Weis, 2008*). To understand the oxidative stress response in green hydra, we searched the differentially expressed genes with the GO terms 'response to oxidative stress', 'oxidation-reduction process' and 'oxidoreductase activity'. In Hv_Sym, contigs for peroxidase, methionine-r-sulfoxide reductase/selenoprotein and glutaredoxin, which are known to be related to oxidative stress response were up-regulated (*Table 1*). On the other hand, some contigs encoding glutathione S-transferase and ascorbate peroxidase were down-regulated in Hv_Sym. In addition, two contigs encoding polycystin were down-regulated in Hv_Sym. Polycystin is an intracellular calcium release channel that is inhibited by ROS (*Montalbetti et al., 2008*) and is also down-regulated in a different strain of symbiotic green hydra (*Ishikawa et al., 2016*). In addition, six contigs encoding metalloproteinases showed differential expression between Hv_Sym and Hv_Apo. Although metalloproteinases have many functions such as cleavage of cell surface proteins and remodeling of the extracellular matrix, in a previous study they also were found to play a role in the oxidative stress response (*Császár et al., 2009*). A key antioxidant in the oxidative stress response in symbiotic cnidarians turns out to be glutathione (*Sunagawa et al., 2009*; *Meyer and Weis, 2012*). The gene encoding glutathione S-transferase was previously observed to be downregulated in corals, sea anemones, different strains of green hydra and *Paramecium* (*Kodama et al., 2014*; *Lehnert et al., 2014*; *Ishikawa et al., 2016*; *Mohamed et al., 2016*). Our study supports this view (*Table 1*) and may point to a conserved feature of oxidative stress response in algal-animal symbiosis.

Previous studies have suggested that during establishment of coral–algal symbiosis the host immune response may be partially suppressed (Weis et al., 2008; *Mohamed et al., 2016*). Our observations in *Hydra* together with previous findings in corals indicate that regulation of symbiosis by innate immunity pathways indeed may be a general feature of cnidarian symbiosis. Among the differentially expressed contigs we identified a number of genes involved in innate immunity and apoptosis. Pattern recognition receptors (PRRs) and the downstream innate immunity and apoptosis pathways are thought to play important roles in various symbiotic interactions including cnidarian-dinoflagellate symbiosis (*Davy et al., 2012*). In Hv_Sym we found two up-regulated contigs that contain a Toll/interleukin-1 receptor (TIR) domain (*Table 1*). TIR is a known PRR that is inserted in the host cell membrane and plays an important role in the innate immune system by specifically recognizing microbial-associated molecular patterns, such as flagellin, lipopolysaccharide (LPS) and peptidoglycan (*Hoving et al., 2014*). Furthermore, we found one up-regulated contig with similarity to a mannose receptor gene with C-type lectin domain (*Table 1*). This is worth noting since C-type lectin receptors bind carbohydrates and some of them are known to function as PRRs. Host lectin-algal glycan interactions have been proposed to be involved in infection and recognition of symbionts in some cnidarians including green hydra, sea anemones and corals (*Meints and Pardy, 1980*; *Lin et al., 2000*; *Wood-Charlson et al., 2006*). Interestingly, up-regulation of C-type lectin genes was also observed during onset of cnidarian–dinoflagellate symbiosis (*Grasso et al., 2008*; *Schwarz et al., 2008*; *Sunagawa et al., 2009*; *Mohamed et al., 2016*).

Furthermore, contigs encoding chitinase enzymes also were differentially expressed between Hv_Sym and Hv_Apo (*Table 1*). Chitinases are involved in degradation of chitin, which is a component of the exoskeleton of arthropods and the cell wall of fungi, bacteria and some *Chlorella* algae (*Kapaun and Reisser, 1995*), and also might play a role in host-defense systems for pathogens which have chitinous cell wall. Chitinases are classified into two glycoside hydrolase families, GH18 and GH19, with different structures and catalytic mechanisms. In Hv_Sym two contigs encoding GH18 chitinases were up-regulated, while one contig encoding a GH19 chitinase was down-regulated, suggesting that the enzymes involved in chitin degradation are sensitive to the presence or absence of symbiotic *Chlorella*.

To narrow down the number of genes specifically affected by the presence of the native symbiont *Chlorella* A99, we identified 12 contigs that are differentially expressed in symbiosis with *Chlorella* A99, but not in presence of foreign *Chlorella* NC64A (*Figure 1C* A99-specific). Independent qPCR confirmed the differential expression pattern for 10 of these genes (*Table 2*). The genes up-

**Table 2.** List of genes differentially expressed in Hv_Sym compared to both Hv_Apo and Hv_NC64A ('A99-specific')
Fold change of expression level determined by microarray analysis and qPCR analysis

**Hv_Sym > Hv_Apo, Hv_NC64A**

| Probe name (gene ID) | Microarray | | qPCR | | Gene annotation | InterProScan |
|---|---|---|---|---|---|---|
| | Sym/Apo | Sym/NC64A | Sym/Apo | Sym/NC64A | | |
| rc_13579 | 12.8 | 4.0 | 11.2 | 4.0 | (Hydra specific) | |
| rc_12891 | 9.0 | 2.9 | 14.6 | 6.9 | (Hydra viridis specific) | |
| 27417 | 4.5 | 4.8 | 3.0 | 3.0 | | IPR009786 Spot_14 |
| rc_26218 | 3.3 | 2.4 | 2.5 | 2.3 | sodium-dependent phosphate transport protein | PTHR10010 Sodium-dependent phosphate transport protein 2C |
| 1046 | 3.1 | 2.1 | 2.2 | 1.6 | glutamine synthetase | |

**Hv_Sym < Hv_Apo, Hv_NC64A**

| Probe name (gene ID) | Microarray | | qPCR | | Gene Annotation | InterProScan |
|---|---|---|---|---|---|---|
| | Apo/Sym | NC64A/Sym | Apo/Sym | NC64A/Sym | | |
| NPNHRC_26859 | 83.2 | 9.7 | ∞ | ∞ | (Hydra viridis specific) | |
| RC_FVQRUGK01AXSJ | 13.7 | 2.6 | 2.1 | 1.5 | acetoacetyl-CoA synthetase | |
| rc_14793 | 7.2 | 4.1 | 9.4 | 4.8 | 2-isopropylmalate synthase | IPR013785 Aldolase_TIM, |
| FV81RT002HT2FL | 2.8 | 2.0 | 3.1 | 1.8 | histidine ammonia-lyase | IPR001106 Aromatic_Lyase IPR008948 L-Aspartase-like |
| NPNHRC_12201 | 2.7glutamate | 2.3 | 2.6 | 2.5 | (Hydra viridis specific) | |

DOI: https://doi.org/10.7554/eLife.35122.009
The following source data available for Table 2:
Source data 1. Expression level of 'A99-specific' genes and 'Symbiosis related' genes examined by microarray and qPCR.
DOI: https://doi.org/10.7554/eLife.35122.010

regulated by the presence of the symbiont encode a Spot_14 protein, a glutamine synthetase (GS) and a sodium-dependent phosphate (Na/Pi) transport protein in addition to a *H. viridissima* specific gene (rc_12891: *Sym-1*) and a *Hydra* genus specific gene (rc_13570: *Sym-2*) (*Table 2*). *Hydra* genes down-regulated by the presence of *Chlorella* A99 were two *H. viridissima*-specific genes and three metabolic genes encoding histidine ammonia-lyase, acetoacetyl-CoA synthetase and 2-isopropylmalate synthase (*Table 2*). Of the up-regulated genes, Spot_14 is described as thyroid hormone-responsive spot 14 protein reported to be induced by dietary carbohydrates and glucose in mammals (*Tao and Towle, 1986*; *Brown et al., 1997*). Na/Pi transport protein is a membrane transporter actively transporting phosphate into cells (*Murer and Biber, 1996*). GS plays an essential role in the metabolism of nitrogen by catalyzing the reaction between glutamate and ammonia to form glutamine (*Liaw et al., 1995*). Interestingly, out of the three GS genes *H. viridissima* contains only *GS-1* was found to be up-regulated by the presence of the symbiont (*Figure 1—figure supplement 3*). The discovery of these transcriptional responses points to an intimate metabolic exchange between the partners in a species-specific manner.

To better understand the specificity of *Hydra*'s response to the presence of the foreign symbiont, we also identified the genes differentially expressed in *Hydra* polyps hosting a non-native *Chlorella* NC64A (Hv_NC64A) compared to both polyps hosting the obligate symbiont *Chlorella* A99 (Hv_A99) and aposymbiotic Hydra (Hv_Apo). We found 19 contigs that were up-regulated and 45 contigs that were down-regulated in presence of NC64A, which strikingly did not include any genes related to immunity or oxidative stress response (*Supplementary file 1*). Instead, the differentially expressed contigs showed similarity to methylase genes involved in ubiquinone menaquinone biosynthesis and secondary metabolite synthesis such as n-(5-amino-5-carboxypentanoyl)-l-cysteinyl-d-valine synthase and non-ribosomal peptide synthase. Four differentially expressed contigs specifically up-regulated in Hv_NC64A encoded ubiquitin carboxyl-terminal hydrolases, (*Table 3*).

**Table 3.** List of annotated genes up-regulated in Hv_NC64A compared to Hv_Sym

| Probename | Hv_NC64A/ Hv_Sym | Hv_Apo/ Hv_Sym | Hv_Sym_sexy/ Hv_Sym | Blast2GO description |
|---|---|---|---|---|
| rc_1623 | 4.57 | 1.64 | 5.98 | methylase involved in ubiquinone menaquinone biosynthesis |
| 28947 | 3.52 | 1.59 | 0.63 | non-ribosomal peptide synthetase |
| 1353 | 3.13 | 1.63 | 0.10 | nuclear protein set |
| 14347 | 2.69 | 2.40 | 0.54 | n-(5-amino-5-carboxypentanoyl)-l-cysteinyl-d-valine synthase |
| SSH_397 | 2.67 | 2.39 | 0.50 | n-(5-amino-5-carboxypentanoyl)-l-cysteinyl-d-valine synthase |
| RC_FWZAEML01C7BP | 2.28 | 0.82 | 0.41 | ubiquitin carboxyl-terminal hydrolase family protein |
| RC_FVQRUGK01EOXS | 2.25 | 1.52 | 0.53 | ubiquitin carboxyl-terminal hydrolase family protein |
| rc_11710 | 2.15 | 1.26 | 0.31 | ubiquitin carboxyl-terminal hydrolase family protein |
| 1677 | 2.10 | 1.19 | 0.38 | ubiquitin carboxyl-terminal hydrolase family protein |
| rc_363 | 2.21 | 1.04 | 0.76 | gcc2 and gcc3 family protein |

DOI: https://doi.org/10.7554/eLife.35122.011

## Symbiont-dependent *Hydra* genes are up-regulated by photosynthetic activity of *Chlorella A99*

To test whether photosynthetic activity of the symbiont is required for up-regulation of gene expression, Hv_Sym was either cultured under a standard 12 hr light/dark alternating regime or continuously in the dark for 1 to 4 days prior to RNA extraction (*Figure 2A*). Interestingly, four (*GS1*, *Spot14*, *Na/Pi* and *Sym-1*) of five genes specifically activated by the presence of *Chlorella* A99 showed significant up-regulation when exposed to light (*Figure 2B*), indicating the relevance of photosynthetic activity of *Chlorella*. This up-regulation was strictly dependent on presence of the algae, as in aposymbiotic Hv_Apo the response was absent (*Figure 2B*). On the other hand, symbiosis-regulated *Hydra* genes not specific for *Chlorella* A99 (*Figure 1C* Symbiosis-regulated, *Table 4*) appear to be not up-regulated in a light-dependent manner (*Figure 2—figure supplement 1*). These genes are involved in *Hydra*´s innate immune system (e.g. proteins containing Toll/interleukin-1 receptor domain or Death domain) or in signal transduction (C-type mannose receptor, ephrin receptor, proline-rich transmembrane protein 1, 'protein-kinase, interferon-inducible double stranded RNA dependent inhibitor, repressor of (p58 repressor)'). That particular transcriptional changes observed in *Hydra* rely solely on the photosynthetic activity of *Chlorella* A99 was confirmed by substituting the dark incubation with selective chemical photosynthesis inhibitor DCMU (Dichorophenyl-dimethylurea) (*Vandermeulen et al., 1972*), which resulted in a similar effect (*Figure 2C,D*).

## Symbiont-dependent *Hydra* genes are expressed in endodermal epithelial cells and up-regulated by sugars

To further characterize the symbiont induced *Hydra* genes, we performed whole mount in situ hybridization (*Figure 3A–F*) and quantified transcripts by qPCR using templates from isolated endoderm and ectoderm (*Figure 3—figure supplement 1*), again comparing symbiotic and aposymbiotic polyps (*Figure 3G–I*). The GS-1 gene and the Spot14 gene are expressed both in ectoderm and in endoderm (*Figure 3A,B*) and both genes are strongly up-regulated in the presence of the symbiont (*Figure 3G,H*). In contrast, the Na/Pi gene was expressed only in the endoderm (*Figure 3C*) and there it was strongly up-regulated by the symbiont (*Figure 3I*). Since *Chlorella* sp. A99 colonizes endodermal epithelial cells only, the impact of algae on symbiosis-dependent genes in both the ectodermal and the endodermal layer indicates that photosynthetic products can be transported across these two tissue layers or some signals can be transduced by cell-cell communication.

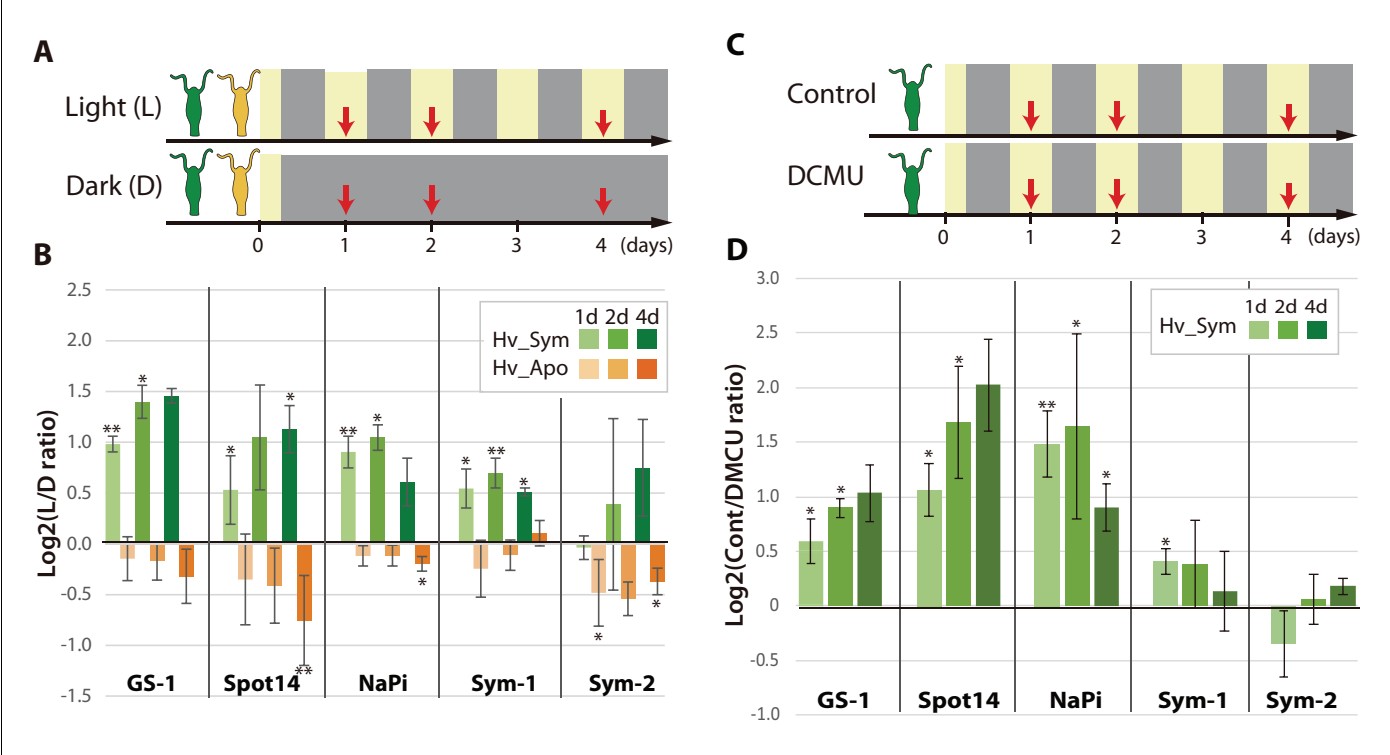

**Figure 2.** Differential expression of *Hydra* genes under influence of *Chlorella* photosynthesis. (A) Sampling scheme. Hv_Sym (green) and Hv_Apo (orange) were cultured under a standard light-dark regime (Light: L) and in continuous darkness (Dark: D), and RNA was extracted from the polyps at the days indicated by red arrows. (B) Expression difference of five A99-specific genes in Hv_Sym (green bars) and Hv_Apo (orange bars) between the light-dark condition and darkness. The vertical axis shows log scale (log2) fold changes of relative expression level in Light over Dark. (C) Sampling scheme of inhibiting photosynthesis. (D) Differential expression of the five A99-specific genes under conditions allowing (Control) or inhibiting photosynthesis (DCMU). The vertical axis shows log scale (log2) fold changes of relative expression level in Control over DCMU treated. T-tests were performed between Light and Dark (B), and DCMU and Control (D). For each biological replicate (n = 3) 50 hydra polyps were used for total RNA extraction. Error bars indicate standard deviation. P-value of t-test, *<0.05, **<0.01.
DOI: https://doi.org/10.7554/eLife.35122.012

The following source data and figure supplements are available for figure 2:

**Figure supplement 1.** Differential expression of symbiosis-dependent *Hydra* genes grown under light/dark condition and in darkness.
DOI: https://doi.org/10.7554/eLife.35122.013

**Figure supplement 1—source data 1.** *Hydra* genes under influence of *Chlorella* photosynthesis examined by qPCR.
DOI: https://doi.org/10.7554/eLife.35122.014

To more closely dissect the nature of the functional interaction between *Hydra* and *Chlorella* and to explore the possibility that maltose released from the algae is involved in A99-specific gene regulation, we cultured aposymbiotic polyps (Hv_Apo) for 2 days in medium containing various concentrations of maltose (*Figure 3J*). Of the five A99 specific genes, GS-1 and the Spot14 gene were up-regulated by maltose in a dose-dependent manner; the Na/Pi gene was only up-regulated in 100 mM maltose and the *Hydra* specific genes Sym-1 and Sym-2 did not show significant changes in expression by exposure to maltose (*Figure 3J*). This provides strong support for previous views that maltose excretion by symbiotic algae contributes to the stabilization of this symbiotic association (*Cernichiari et al., 1969*). When polyps were exposed to glucose instead of maltose, the genes of interest were also transcriptionally activated in a dose-dependent manner, while sucrose had no effect (*Figure 3—figure supplement 2A–D*). Exposure to low concentrations of galactose increased transcriptional activity but at high concentration it did not, indicating a substrate inhibitor effect for this sugar. That the response to glucose is similar or even higher compared to maltose after 6 hr of treatment (*Figure 3—figure supplement 2E*), suggests that *Hydra* cells transform maltose to glucose as a source of energy. In animals including cnidarians, several glucose transporters have been

**Table 4.** List of the genes differentially expressed between Hv_Sym and Hv_Apo
Fold change of expression level determined by microarray analysis and qPCR

Hv_Sym > Hv_Apo

| Probe name (gene ID) | Microarray | qPCR | Gene annotation | InterProScan |
| --- | --- | --- | --- | --- |
| | Sym/Apo | Sym/Apo | | |
| 5168 | 9.3 | 7.4 | | IPR000157 TIR_dom<br>PTHR23097 Tumor necrosis factor receptor superfamily member |
| 6508 | 6.7 | 2.9 | | IPR011029:DEATH-like_dom |
| 11411 | 2.9 | 2.0 | C-type mannose receptor 2 | IPR000742 EG-like_dom<br>IPR001304 C-type_lectin |
| 26108 | 7.2 | 7.2 | ephrin type-A receptor six isoform a | |
| rc_2417 | 5.4 | 3.5 | | IPR000488 Death_domain |
| rc_24563 | 6.1 | 6.7 | Proline-rich transmembrane protein 1 | IPR007593 CD225/Dispanin_fam<br>PTHR14948 NG5 |
| rc_9398 | 6.2 | 5.4 | protein-kinase, interferon-inducible double stranded RNA dependent inhibitor, repressor of (P58 repressor) | PTHR11697 general transcription factor 2-related zinc finger protein |

Hv_Sym < Hv_Apo

| Probe name (gene ID) | Microarray | qPCR | Gene Annotation | InterProScan |
| --- | --- | --- | --- | --- |
| | Apo/Sym | Apo/Sym | | |
| rc_10789 | 2.5 | 3.7 | endoribonuclease Dicer | IPR000999 RNase_III_dom<br>PTHR1495 helicase-related |
| rc_12826 | 3.0 | 2.3 | interferon regulatory factor 1 | IPR001346 Interferon_reg_fact_DNA-bd_dom;<br>IPR011991 WHTH_DNA-bd_dom<br>PTHR11949 interferon regulatory factor |
| rc_8898 | 6.1 | 4.1 | leucine-rich repeat-containing protein 15 isoform b | IPR001611 Leu-rich_rp<br>PTHR24373 Toll-like receptor 9 |
| FV81RT001CSTY | 3.2 | 2.0 | astrocytic phosphoprotein PEA-15 | IPR001875 DED, IPR011029 DEATH-like_dom |
| RSASM_17752 | 4.0 | 2.1 | CD97 antigen isoform two precursor | IPR000832 GPCR_2_secretin-like<br>PTHR12011 vasoactive intestinal polypeptide receptor 2 |

DOI: https://doi.org/10.7554/eLife.35122.015

The following source data available for Table 4:
Source data 1. Expression level of 'Symbiosis related' genes examined by microarray and qPCR.
DOI: https://doi.org/10.7554/eLife.35122.016

identified (*Sproles et al., 2018*), but yet no maltose transporters. This is consistent with the view that maltose produced by the symbiont is digested to glucose in the symbiosome and translocated to the host cytoplasm through glucose transporters.

## The *Chlorella* A99 genome records a symbiotic life style

To better understand the symbiosis between *H. viridissima* and *Chlorella* and to refine our knowledge of the functions that are required in this symbiosis, we sequenced the genome of *Chlorella* sp. strain A99 and compared it to the genomes of other green algae. The genome of *Chlorella* sp. A99 was sequenced to approximately 211-fold coverage, enabling the generation of an assembly comprising a total of 40.9 Mbp (82 scaffolds, N50 = 1.7 Mbp) (*Table 5*). *Chlorella* sp. A99 belongs to the family *Chlorellaceae* (*Figure 4A*) and of the green algae whose genomes have been sequenced it is most closely related to *Chlorella variabilis* NC64A (NC64A) (*Merchant et al., 2007*; *Palenik et al., 2007*; *Worden et al., 2009*; *Blanc et al., 2010*; *Prochnik et al., 2010*; *Blanc et al., 2012*; *Gao et al., 2014*; *Pombert et al., 2014*). The genome size of the total assembly in strain A99 was similar to that of strain NC64A (46.2 Mb) (*Figure 4B*). By k-mer analysis (k-mer = 19), the genome size of A99 was estimated to be 61 Mbp (*Marçais and Kingsford, 2011*). Its GC content of 68%, is the highest among the green algae species recorded (*Figure 4B*). In the A99 genome, 8298 gene

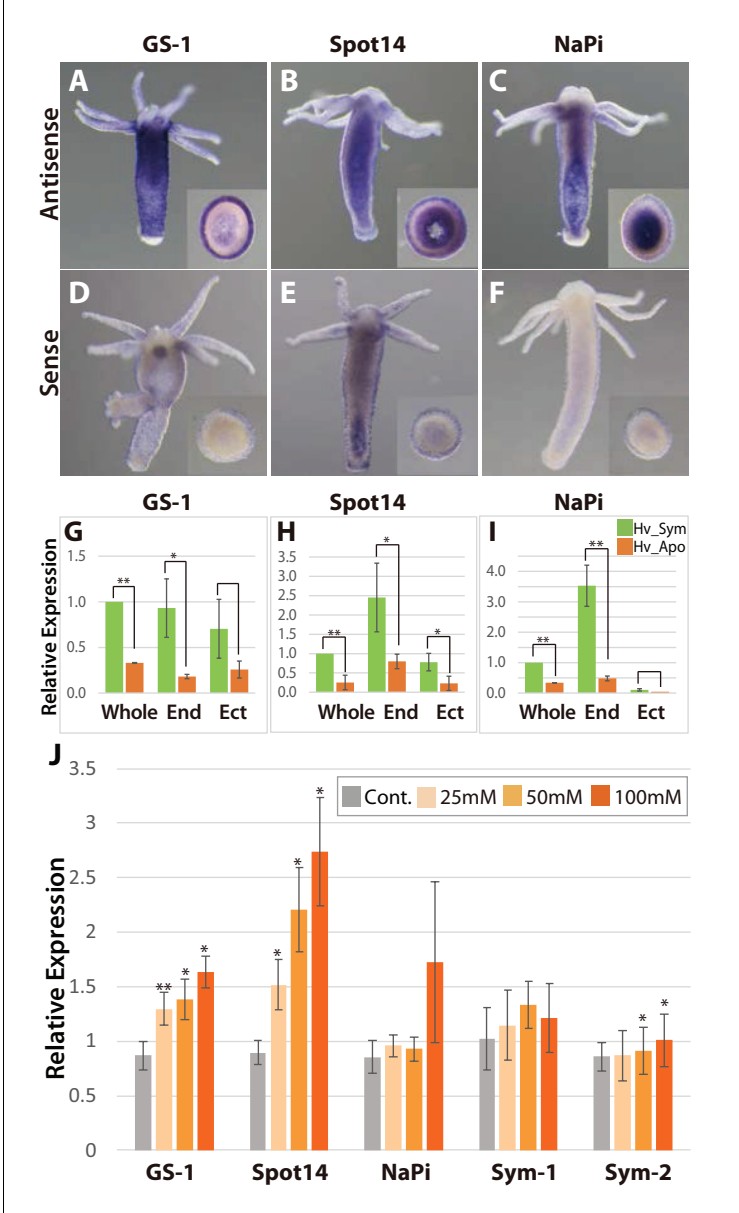

**Figure 3.** Spatial expression patterns of genes coding for glutamine synthetase, Spot 14 and Na/Pi-transporter. (A-F); Whole mount in situ hybridization using antisense (A–C) and sense probes (D-F; negative controls) for glutamine synthetase-1 (GS-1; left), Spot 14 (center) and Na/Pi-transporter (NaPi; right). Inserts show cross sections of the polyp's body. (G–I) Relative expression levels of whole animal (whole), isolated endoderm (End) and isolated ectoderm (Ect) tissue of Hv_Sym (green bars) and Hv_Apo (orange bars). For each biological replicate (n = 3) 10–20 hydra polyps were used for total RNA extraction of endodermal and ectodermal tissue. T-test was performed between Hv_Sym and Hv_apo. Pvalue, *<0.05, **<0.01. (J) Expression change of genes GS-1, Spot14, NaPi, Sym-1 and Sym-2 following exposure to 25, 50 and 100 mM maltose in Hv_Apo. For each biological replicate (n = 3) 50 hydra polyps were used for total RNA extraction The vertical axis shows log scale (log2) fold changes of relative expression level of maltose-treated over the untreated Hv_Apo control. T-test was performed between maltose-treated in each concentration and control (*: p value <0.05) and Kruskal-Wallis test (†: p value <0.05) in the series of 48 hr treatment were performed. Error bars indicate standard deviation.

DOI: https://doi.org/10.7554/eLife.35122.017

The following source data and figure supplements are available for figure 3:

**Source data 1.** Expression change of genes GS-1, Spot14, NaPi, Sym-1 and Sym-2 following exposure to 25, 50 and 100 mM maltose in Hv_Apo examined by qPCR.

DOI: https://doi.org/10.7554/eLife.35122.021

*Figure 3 continued on next page*

*Figure 3 continued*

**Figure supplement 1.** Tissue isolation of green hydra.
DOI: https://doi.org/10.7554/eLife.35122.018
**Figure supplement 2.** Effects of sugars on *Hydra* growth.
DOI: https://doi.org/10.7554/eLife.35122.019
**Figure supplement 2—source data 1.** Effects in presence of maltose, glucose, sucrose and galactose on gene expression of GS-1, Spot14 and NaPi in Hv_Apo examined by qPCR.
DOI: https://doi.org/10.7554/eLife.35122.020

models were predicted. As shown in *Figure 4C*, about 80% of these predicted genes have extensive sequence similarity to plant genes, while 13% so far have no similarity to genes of any other organisms (*Figure 4C*). It is also noteworthy that 7% of the A99 genes are similar to genes of other kingdoms but not to *Hydra*, indicating the absence of gene transfer from *Hydra* to the symbiont genome (*Figure 4C*).

## The *Chlorella* A99 genome provides evidences for extensive nitrogenous amino acid import and an incomplete nitrate assimilation pathway

Several independent lines of evidence demonstrate that nitrogen limitation and amino-acid metabolism have a key role in the *Chlorella–Hydra* symbiosis and that symbiotic *Chlorella* A99 depends on glutamine provided by its host (*Rees, 1986*; *McAuley, 1987a*; *1987b*; *McAuley, 1991*; *Rees, 1991*;*1989*). To identify *Chlorella* candidate factors for the development and maintenance of the symbiotic life style, we therefore used the available genome information to assess genes potentially involved in amino acid transport and the nitrogen metabolic pathway.

When performing a search for the Pfam domain 'Aa_trans' or 'AA_permease' to find amino acid transporter genes in the A99 genome, we discovered numerous genes containing the Aa_trans domain (*Table 6A*). In particular, A99 contains many orthologous genes of amino acid permease 2 and of transmembrane amino acid transporter family protein (solute carrier family 38, sodium-coupled neutral amino acid transporter), as well as NC64A (*Table 6B*, *Supplementary file 2*). Both of these gene products are known to transport neutral amino acids including glutamine. This observation is supporting the view that import of amino acids is an essential feature for the symbiotic way of life of *Chlorella*.

In symbiotic organisms, loss of genes often occurs due to the strictly interdependent relationship (*Ochman and Moran, 2001*; *Wernegreen, 2012*), raising the possibility that *Chlorella* A99 might have lost some essential genes. To test this hypothesis, we searched the *Chlorella* A99 genome for genes conserved across free-living green algae *Coccomyxa subellipsoidea* C169 (C169), *Chlamydomonas reinhardtii* (Cr) and *Volvox carteri* (Vc). In a total of 9851 C169 genes, we found 5701 genes to be conserved in Cr and Vc (*Supplementary file 3*). Of these, 238 genes did not match to any gene models and genomic regions in *Chlorella* A99 and thus were considered as gene loss candidates. Interestingly, within these 238 candidates, genes with the GO terms 'transport' in biological

**Table 5.** Summary of sequence data for assembling *Chlorella* sp. A99 genome sequences

| | | |
|---|---|---|
| Number of reads | 85469010 | |
| Number of reads assembled | 61838513 | |
| Number of bases | 17398635102 | |
| | Scaffolds | Contigs |
| Total length of sequence | 40934037 | 40687875 |
| Total number of sequences | 82 | 7455 |
| Maximum length of sequence | 4003385 | 171868 |
| N50 | 1727419 | 12747 |
| GC contents (%) | 68.07% | 69.95% |

DOI: https://doi.org/10.7554/eLife.35122.023

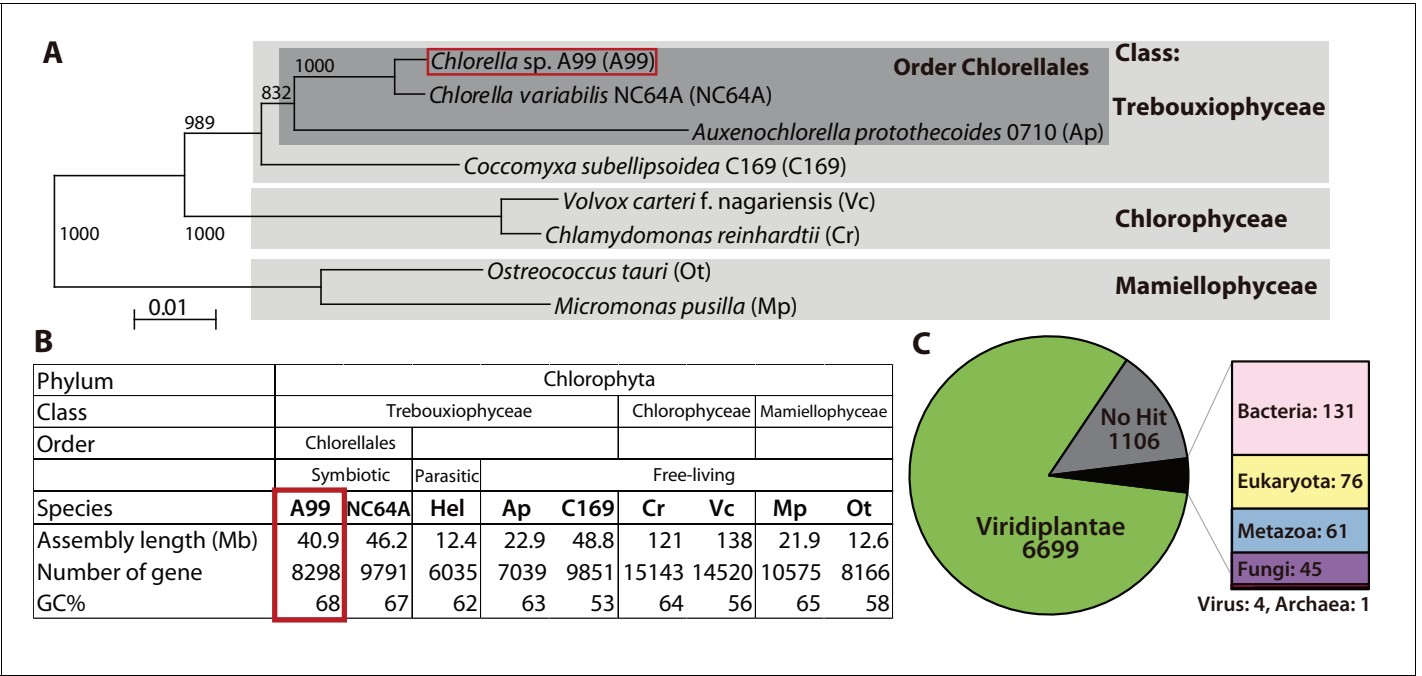

**Figure 4.** Comparison of key features deduced from the *Chlorella* A99 genome with other green algae. (**A**) Phylogenetic tree of eight genome sequenced chlorophyte green algae including *Chlorella* sp. A99. The NJ tree is based on sequences of the 18S rRNA gene, ITS1, 5.8S rRNA gene, ITS2 and 28S rRNA gene. (**B**) Genomic features and taxonomy of the sequenced chlorophyte green algae. Hel: *Helicosporidium* sp. ATCC50920. (**C**) The proportion of similarity of *Chlorella* A99 gene models to those of other organisms.

DOI: https://doi.org/10.7554/eLife.35122.022

process and 'transporter activity' in molecular function were overrepresented (*Figure 5*). In particular, the 28 genes annotated to these GO terms encoded nitrate transporter, urea transporter and molybdate transporter, which are known to be involved in nitrogen metabolism (*Table 7*). Beside ammonium, nitrate and urea are major nitrogen sources for plants, whereas molybdate is a co-factor of the nitrate reductase, an important enzyme in the nitrate assimilation pathway. These transporter genes are conserved across green algae including *Chlorella* NC64A (*Sanz-Luque et al., 2015*; *Gao et al., 2014*) and appear to be lost in the *Chlorella* A99 genome.

In nitrogen assimilation processes, plants usually take up nitrogen in the form of nitrate ($NO_3^-$) via nitrate transporters (NRTs) or as ammonium ($NH_4^+$) via ammonium transporters (AMT) (*Figure 6A*). In higher plants, two types of nitrate transporters, NRT1 and NRT2, have been identified (*Krapp et al., 2014*). Some NRT2 require nitrate assimilation-related component 2 (NAR2) to be

**Table 6.** Amino acid transporter genes in *Chlorella* sp. A99 (A99), *Chlorella variabilis NC64A* (NC64A), *Coccomyxa subellipsoidea* C-169 (C169), *Volvox carteri* (Vc), *Micromonas pusilla* (Mp) and *Ostreococcus tauri* (Ot) and *Chlamydomonas reinhardtii* (Cr)

A. The number of Pfam domains related to amino acids transport

| Pfam domain name | A99 | NC64A | c169 | Cr | Vc | Mp | Ot |
|---|---|---|---|---|---|---|---|
| Aa_trans | 30 | 38 | 21 | 9 | 7 | 9 | 8 |
| AA_permease | 4 | 6 | 15 | 5 | 6 | 1 | 1 |

B. Ortholog groups including Aa_trans domain containing genes overrepresented in symbiotic *Chlorella*

| Ortholog group ID: Gene annotation | A99 | NC64A | c169 | Cr | Vc | Mp | Ot |
|---|---|---|---|---|---|---|---|
| OG0000040: amino acid permease 2 | 12 | 12 | 6 | 3 | 1 | 0 | 0 |
| OG0000324: transmembrane amino acid transporter family protein (solute carrier family 38, sodium-coupled neutral amino acid transporter) | 6 | 7 | 1 | 2 | 1 | 0 | 0 |

DOI: https://doi.org/10.7554/eLife.35122.024

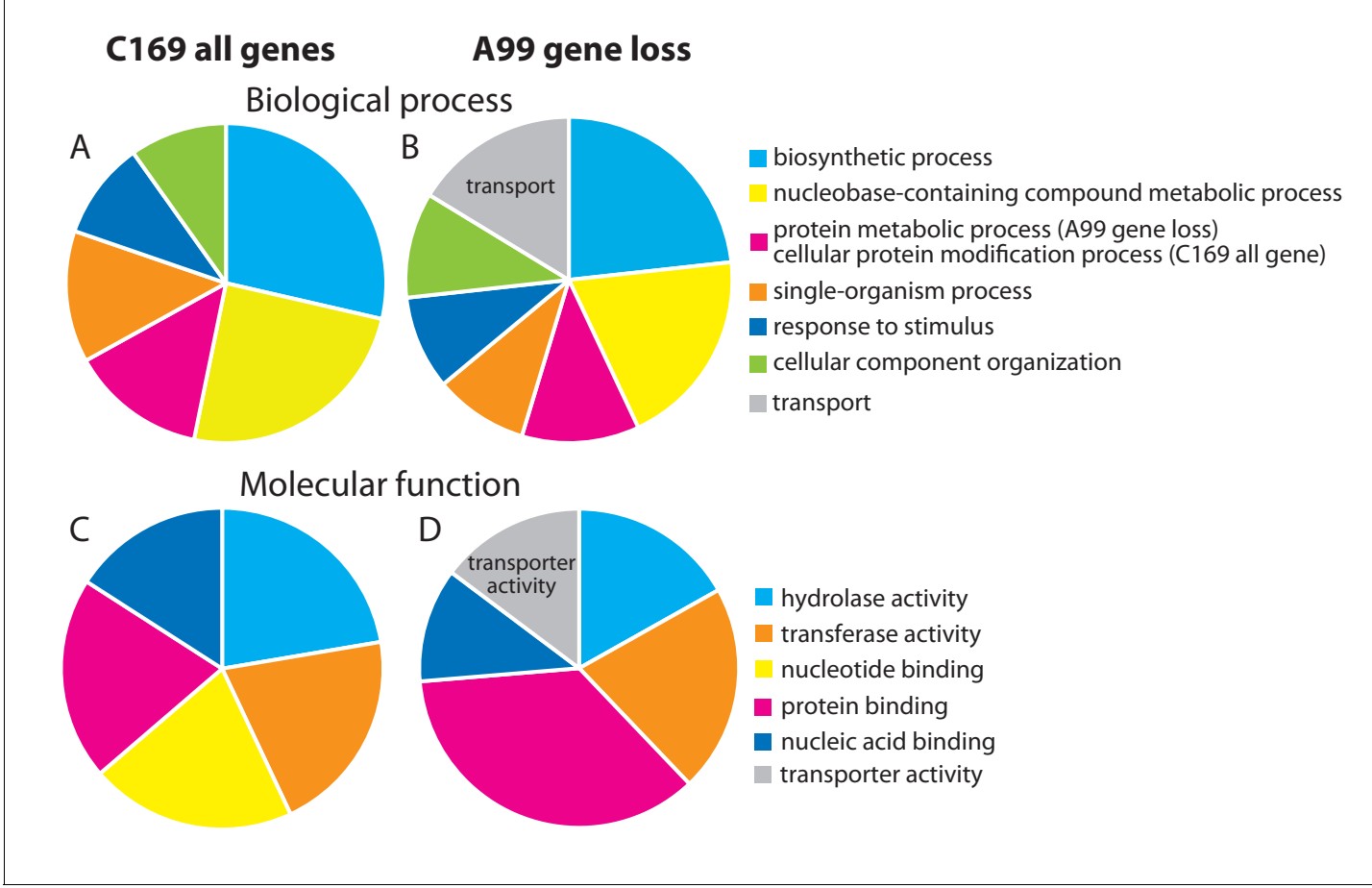

**Figure 5.** Genes missing in the genome of *Chlorella* A99. Functional categorization of genes present in *Coccomyxa subellipsoidea* C169 (**A, C**) and genes missing in *Chlorella* A99 (**B, D**) by GO terms using Bast2GO. Multilevel pie charts show enrichment of GO' Biological Process' terms (**A, B**) and GO 'Molecular Function' terms (**C, D**) on the lowest level, which cover at least 10% of the total amount of annotated sequences.
DOI: https://doi.org/10.7554/eLife.35122.025

The following source data is available for figure 5:

**Source data 1.** Functional categorization of genes present in *Coccomyxa subellipsoidea* C169 (C169_all) and genes missing in Chlorella A99 (A99 gene loss) by GO terms' Biological Process' terms and 'Molecular Function' on the lowest level, which cover at least 10% of the total amount of annotated sequences.
DOI: https://doi.org/10.7554/eLife.35122.026

functional (*Quesada et al., 1994*). $NO_3^-$ is reduced to nitrite by nitrate reductase (NR), $NO_2^-$ is transported to the chloroplast by nitrate assimilation-related component1 (NAR1), and $NO_2^-$ is reduced to $NH_4^+$ by nitrite reductase (NiR). $NH_4^+$ is incorporated into glutamine (Gln) by glutamine synthetase (GS), and Gln is incorporated into glutamate (Glu) by NADH-dependent glutamine amide-2-oxoglutarate aminotransferase (GOGAT), also known as glutamate synthase. This pathway is highly conserved among plants and all of its major components, including NRT1 and NRT2, NAR1 and NAR2, NR, NiR, AMT, GOGAT and GS, are present in the 10 green algae species that have been genome-sequenced so far (with the exception of NRT1, which is absent in *Micromonas pusilla*) (*Sanz-Luque et al., 2015*). In *Symbiodinium,* the photosynthetic symbiont of marine invertebrates, all these components of the nitrogen assimilation pathway were also observed (*Supplementary file 4*) (*Shoguchi et al., 2013*; *Lin et al., 2015*; *Aranda et al., 2016*; *Sproles et al., 2018*).

Based on the annotation by Sanz-Luque et al. (*Sanz-Luque et al., 2015*), we searched these nitrogen assimilation genes in the *Chlorella* A99 genome, using ortholog grouping and a reciprocal BLAST search using the protein sequences from other green algae (*Figure 6B*, *Supplementary file 5*). As expected, the *Chlorella* A99 genome contains many homologues of the genes involved in

**Table 7.** List of *Coccomyxa subellipsoidea* C169 (C169) genes, which are present in *Chlamydomonas reinhardtii* and *Volvox carteri*, but missing in the genome of *Chlorella* A99

| UniProt ID in C169 | Description |
| --- | --- |
| F1DPL8_9CHLO | ATP synthase F0 subunit 6 (mitochondrion) |
| F1DPL7_9CHLO | cytochrome c oxidase subunit 3 (mitochondrion) |
| I0YZU4_9CHLO | equilibrative nucleoside transporter 1 |
| I0Z311_9CHLO | equilibrative nucleoside transporter family |
| I0YZC9_9CHLO | high affinity nitrate transporter |
| I0Z2L2_9CHLO | hypothetical protein COCSUDRAFT_28432 |
| I0YJ99_9CHLO | hypothetical protein COCSUDRAFT_34498 |
| I0YKQ1_9CHLO | hypothetical protein COCSUDRAFT_45098 |
| I0YYD3_9CHLO | hypothetical protein COCSUDRAFT_65897 |
| I0YYP5_9CHLO | importin-4 isoform X1 |
| I0YQQ1_9CHLO | low-CO2-inducible membrane |
| I0YJD4_9CHLO | MFS transporter |
| I0YTY0_9CHLO | molybdate transporter 2 |
| F1DPM0_9CHLO | NADH dehydrogenase subunit 3 (mitochondrion) |
| F1DPM4_9CHLO | NADH dehydrogenase subunit 6 (mitochondrion) |
| F1DPM8_9CHLO | NADH dehydrogenase subunit 9 (mitochondrion) |
| I0Z357_9CHLO | plasma membrane phosphate transporter Pho87 |
| I0Z9Y1_9CHLO | pre translocase subunit |
| I0YPT2_9CHLO | transcription and mRNA export factor ENY2-like |
| I0Z976_9CHLO | transport SEC23 |
| I0Z3Q6_9CHLO | tyrosine-specific transport -like isoform X1 |
| I0YXU9_9CHLO | urea active transporter |
| I0YRT0_9CHLO | urea active transporter |
| I0YRL4_9CHLO | urea-proton symporter DUR3 |
| I0YUF9_9CHLO | urea-proton symporter DUR3 |
| I0YJS6_9CHLO | urea-proton symporter DUR3 |
| I0YQ78_9CHLO | urea-proton symporter DUR3-like |
| I0YIH7_9CHLO | Zip-domain-containing protein |

DOI: https://doi.org/10.7554/eLife.35122.027

nitrogen assimilation in plants including genes encoding NRT1, NAR1, NR, AMT, GS and GOGAT (*Figure 6B*). Intriguingly, our systematic searches failed to identify representative genes for NRT2, NAR2 and NiR in the *Chlorella* A99 genome (*Figure 6B*). We confirmed the absence of the NRT2 and NiR genes by PCR using primers designed for the conserved regions of these genes and which failed to produce a product with genomic DNA as a template (*Figure 6—figure supplement 1*). Due to the weak sequence conservation of the NAR2 gene in the three algae genomes, PCR of that gene was not performed. Taken together, our observations indicate that *Chlorella* A99 algae appears to lack NRT2, NAR2 and NiR.

Since in many fungi, cyanobacteria and algae species, nitrate assimilation genes are known to act in concert and a gene cluster of NR and NiR genes is conserved between different green algae (*Sanz-Luque et al., 2015*), we next investigated the level of genomic clustering of the nitrate assimilation pathway genes in the *Chlorella* genome. Comparing the genomes of NC64A and C169 revealed the presence of a cluster of NR and NiR genes (*Figure 6C*). In NC64A, two NRT2 genes, together with genes for NAR2, NR and NiR are clustered on scaffold 21. In C169, one of the NR genes and NiR are clustered together, whereas the second NR gene is separate. Interestingly, analysis of the sequences around the NR gene in the *Chlorella* A99 genome provided no evidence for the

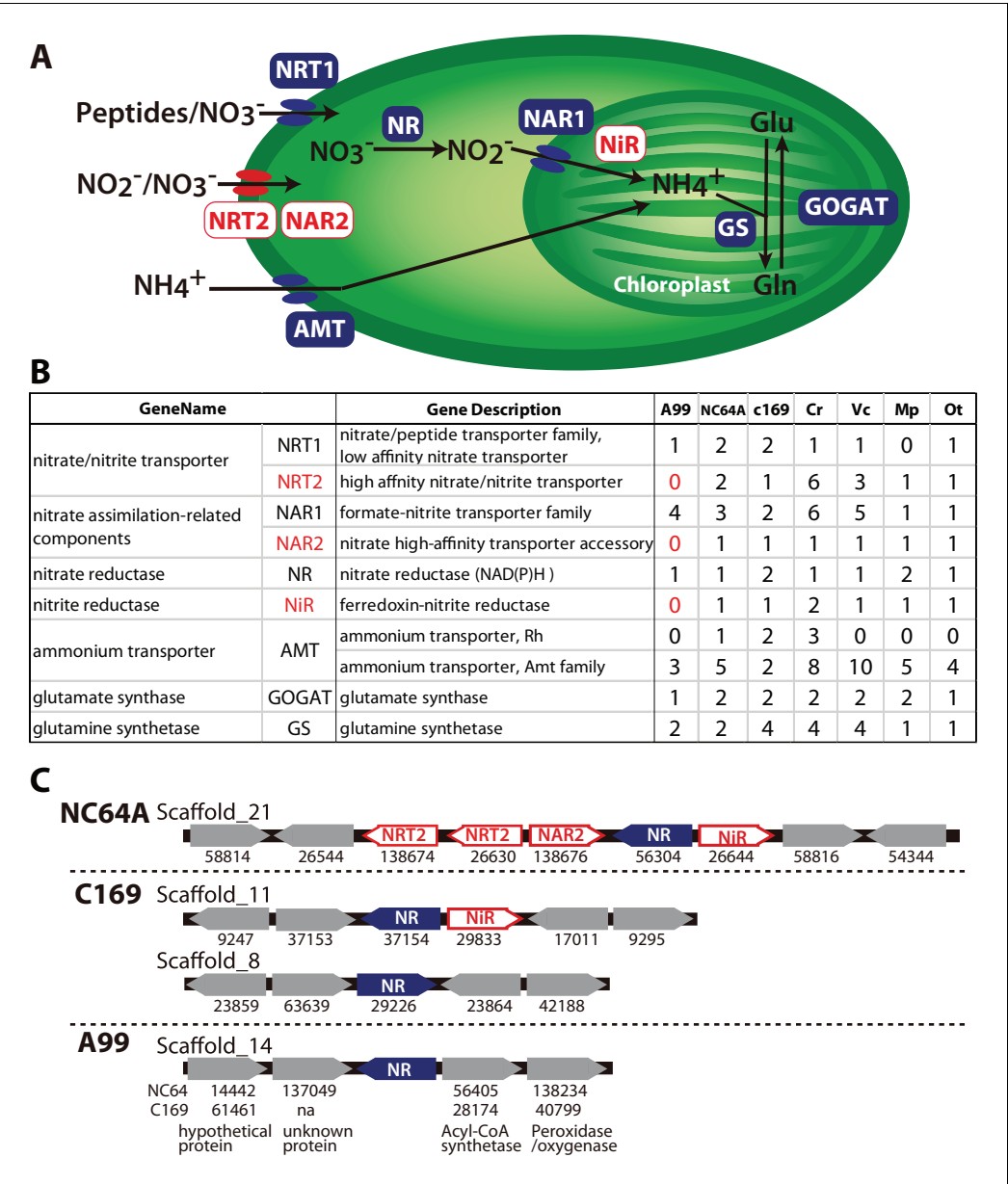

**Figure 6.** Nitrogen assimilation pathways in *Chlorella* A99. (**A**) Schematic diagram of the nitrogen assimilation pathway in plants showing the function of nitrate transporters NRT1 (peptides/nitrate transporter) and NRT2 (nitrate/nitrite transporter), nitrate assimilation-related components NAR1 and NAR2, nitrate reductase NR, nitrite reductase NiR, ammonium transporter AMT, glutamate synthetase GOGAT and glutamine synthetase GS. Genes shown in red boxes (NRT2, NAR2 and NiR) were not found in the *Chlorella* sp. A99 genome. (**B**) Table showing the number of nitrogen assimilation genes in *Chlorella* sp. A99 (**A99**), *Chlorella variabilis* NC64A (NC64A), *Coccomyxa subellipsoidea* C169 (C169), *Volvox carteri* f. *nagariensis* (Vc), *Chlamydomonas reinhardtii* (Cr), *Ostreococcus tauri* (Ot) *and Micromonas pusilla* (Mp). (**C**) Gene clusters of nitrate assimilation genes around the shared NR genes (blue) in the genomes of NC64A, C169 and A99. Red boxes show nitrate assimilation genes absent in A99 and gray boxes depict other genes. Numbers below the boxes are JGI protein IDs of NC64A and C169. Numbers below the genes of A99 are JGI protein IDs of the best hit genes in NC64A and C169 and their gene name.

DOI: https://doi.org/10.7554/eLife.35122.028

The following figure supplement is available for figure 6:

**Figure supplement 1.** PCR of nitrate assimilation genes.
DOI: https://doi.org/10.7554/eLife.35122.029

presence of a co-localized NiR gene or any other nitrate assimilation genes, nor any conserved gene synteny to NC64A and C169 (*Figure 6C*). Therefore, our comparative genomic analyses points to an incomplete and scattered nitrogen metabolic pathway in symbiotic *Chlorella* A99, which lacks essential transporters and enzymes for nitrate assimilation as well as the clustered structure of nitrate assimilation genes.

## Supplementing the medium with glutamine allows temporary in vitro growth of symbiotic *Chlorella* A99

The absence of genes essential for nitrate assimilation in the *Chlorella* A99 genome (*Figure 6*) is consistent with its inability to grow outside the *Hydra* host cell (*Habetha and Bosch, 2005*) and indicates that *Chlorella* symbionts are dependent on metabolites provided by their host. We hypothesized that *Chlorella* is unable to use nitrite and ammonium as a nitrogen source, and that it relies on *Hydra* assimilating ammonium to glutamine to serve as the nitrogen source. To test this hypothesis and to examine utilization of nitrogen compounds of A99, we isolated *Chlorella* A99 from Hv_Sym and cultivated it in vitro using modified bold basal medium (BBM) (*Nichols and Bold, 1965*) containing the same amount of nitrogen in the form of $NO_3^-$, $NH_4^+$, Gln or casamino acids (*Figure 7*). As controls, *Chlorella variabilis* NC64A (NC64A) isolated from Hv_NC64A and free-living C169 were used. To confirm that the cultured A99 is not contamination, we amplified and sequenced the genomic region of the 18S rRNA gene by PCR (*Figure 7—figure supplement 1*) and checked this against the genomic sequence of A99. Kamako et al. reported that free-living alga *Chlorella vulgaris* Beijerinck var. *vulgaris* grow in media containing only inorganic nitrogen compounds as well as in media containing casamino acids as a nitrogen source, while NC64A required amino acids for growth (*Kamako et al., 2005*). Consistent with these observations, C169 grew in all tested media and NC64A grew in media containing casamino acids and Gln, although its growth rate was quite low in presence of $NH_4^+$ and $NO_3^-$ (*Figure 7*). Remarkably, *Chlorella* A99 increased in cell number for up to 8 days in media containing casamino acids and Gln (*Figure 7*). Similar to NC64A, A99 did not grow in presence of $NH_4^+$ and $NO_3^-$. The growth rates of both A99 and NC64A were higher in medium containing a mixture of amino acids (casamino acids) than the single amino acid Gln. In contrast to NC64A, A99 could not be cultivated permanently in casamino acids or glutamine supplemented medium, indicating that additional growth factors are necessary to maintain in

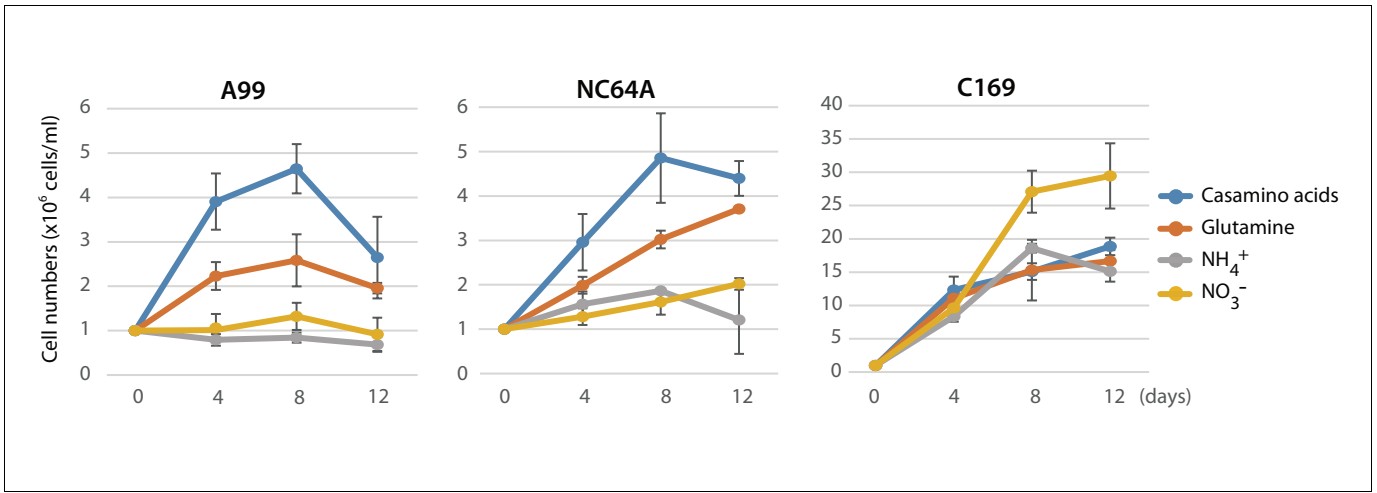

**Figure 7.** Growth of green algae in presence of various nitrogen sources. The growth rate of *Chlorella* A99 (A99), *Chlorella variabilis* NC64A (NC64A) and *Coccomyxa subellipsoidea* C-169 (C169) by in vitro culture was assessed for different nitrogen sources with casamino acids (blue), glutamine (orange), ammonium (gray) and nitrate (yellow). Mean number of algae per ml were determined at 4, 8, 12 days after inoculation with $10^6$ cell/ml. Error bars indicate standard deviation.

DOI: https://doi.org/10.7554/eLife.35122.030

The following figure supplement is available for figure 7:

**Figure supplement 1.** PCR of 18S rRNA genes in cultured algae.

DOI: https://doi.org/10.7554/eLife.35122.031

vitro growth of this obligate symbiont. Thus, although in vitro growth of A99 can be promoted by adding Glu and amino acids to the medium, A99 cannot be cultured permanently in this enriched medium, indicating that other host derived factors remain to be uncovered.

## Discussion

### Metabolic co-dependence in *Hydra-Chlorella* symbiosis

Sequencing of the *Chlorella* A99 genome in combination with the transcriptome analyses of symbiotic, aposymbiotic and NC64A-infected *H. viridissima* polyps has enabled the identification of genes with specific functions in this symbiotic partnership. The *Hydra-Chlorella* symbiosis links carbohydrate supply from the photosynthetic symbiont to glutamine synthesis by the host. Characteristics of the symbiont genome obviously reflect its adaptation to this way of life, including an increase in amino acid transporters and degeneration of the nitrate assimilation pathway. This conclusion is based on six observations: (i) Expression of some genes including GS-1, Spot 14 and NaPi is specifically up-regulated in the presence of *Chlorella* A99 (*Figure 1C*, *Table 2*), and (ii) they are induced by both, photosynthetic activity of *Chlorella* and by supplying exogenous maltose or glucose (*Figures 2* and *3J*, *Figure 3—figure supplement 2*). Maltose produced by the symbiont is likely to be digested to glucose in symbiosome and translocated to the host cytoplasm through glucose transporters (*Figure 8A*). Upregulation of a GLUT8 gene in the symbiotic state of green hydra may reflect activation of sugar transport (*Table 1*). These results indicate that maltose release by photosynthesis of the symbiont enhances nutrition supply including glutamine by the host (*Figure 8A*). (iii) Symbiotic *Chlorella* A99 cannot be cultivated in vitro in medium containing a single inorganic nitrogen source (*Figure 7*). Since medium containing glutamine supports in vitro growth of A99, this organism appears to depend on glutamine provided by the *Hydra* host. (iv) The genome of *Chlorella* A99 contains multiple amino acid transporter genes (*Table 6*), but lacks genes involved in nitrate assimilation (*Figure 6*), pointing to amino acids as main source of nitrogen and a degenerated nitrate assimilation pathway. As for ammonium, which is one of the main nitrogen sources in plants, previous studies have reported the inability of symbiotic algae to take up ammonium because of the low perialgal pH (pH 4–5) that stimulates maltose release (*Douglas and Smith, 1984*; *Rees, 1989*; *McAuley, 1991*; *Dorling et al., 1997*). Since *Chlorella* apparently cannot use nitrite and ammonium as a nitrogen source, it seems that *Hydra* has to assimilate ammonium to glutamine and provides it to *Chlorella* A99 (*Figure 8A*).

(v) While polyps with native symbiont *Chlorella* A99 grew faster than aposymbiotic ones, symbiosis with foreign algae NC64A had no effect on the growth of polyps at all (*Figure 1B*). (vi) *Hydra* endodermal epithelial cells host significantly fewer NC64A algae than A99 (*Figure 1—figure supplement 1*) providing additional support for the view of a tightly regulated codependent partnership in which exchange of nutrients appears to be the primary driving force. Previous studies have reported that symbiotic *Chlorella* in green hydra releases significantly larger amounts of maltose than NC64A (*Mews and Smith, 1982*; *Rees, 1989*). In addition, Rees reported that *Hydra* polyps containing high maltose releasing algae had a high GS activity, whereas aposymbiotic *Hydra* or *Hydra* with a low maltose releasing algae had lower GS activity (*Rees, 1986*). Although the underlying mechanism of how maltose secretion and transportation from *Chlorella* is regulated is still unclear, the amount of maltose released by the symbiont could be an important symbiont-derived driver or stabilizer of the *Hydra–Chlorella* symbiosis.

### More general lessons for animal-algal symbiosis

Transcriptome comparison between symbiotic and aposymbiotic *H. viridissima* demonstrated that symbiosis-regulated genes are involved in oxidative stress response and innate immunity. The fact that PRRs and apoptosis-related genes, are also differentially expressed in a number of other symbiotic cnidarians (*Table 1*), suggests innate immunity as conserved mechanism involved in controlling the development and maintenance of stable symbiotic interactions. Furthermore, the exchange of nitrogenous compounds and photosynthetic products between host and symbiont observed here in the *Hydra-Chlorella* symbiosis is also observed in marine invertebrates such as corals, sea anemones and giant clams associated with *Symbiodinium* algae (*Figure 8B,C*). Despite these similarities, however, there are also conspicuous differences among symbiotic cnidarians in particular with respect to

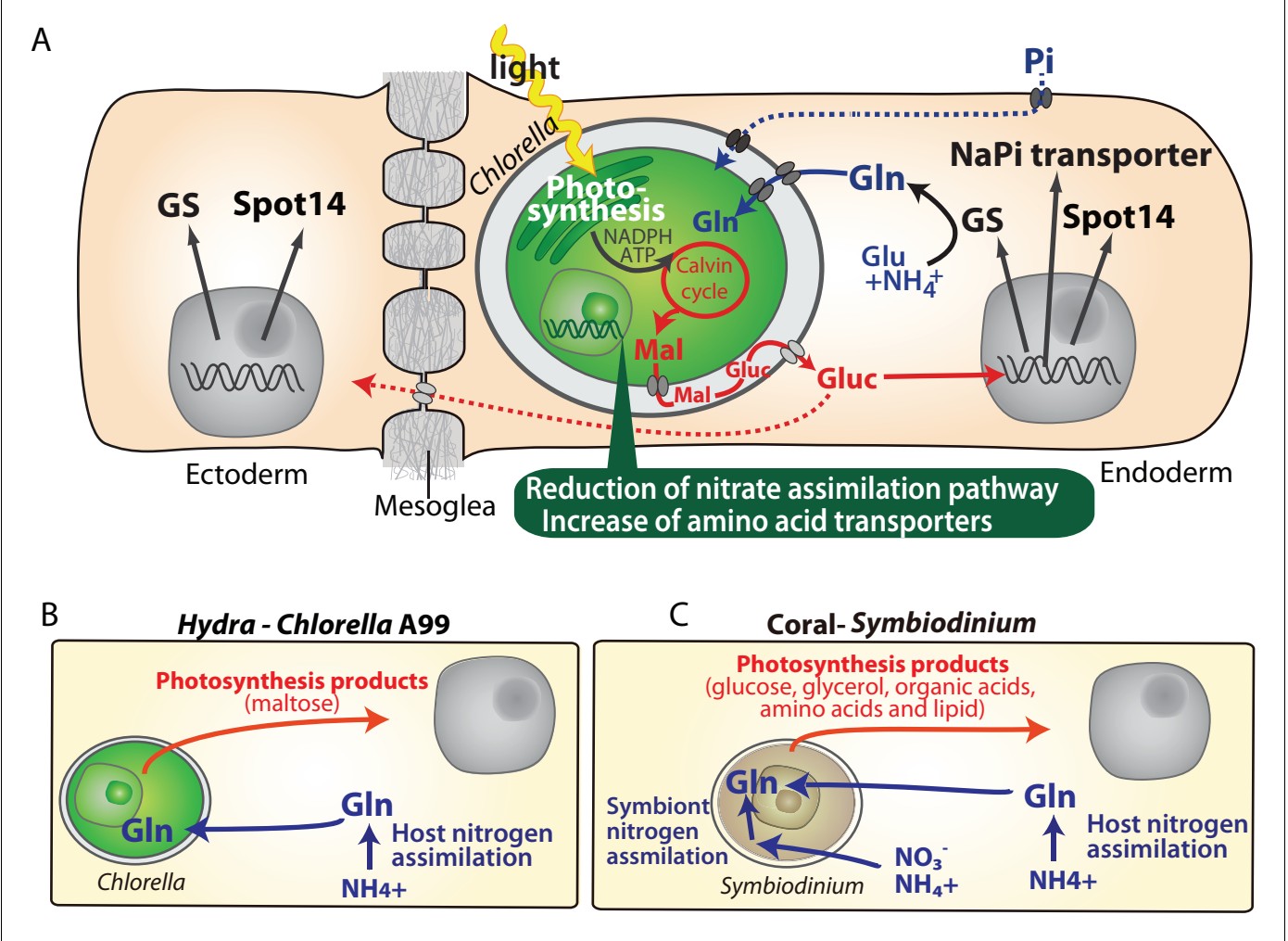

**Figure 8.** Molecular interactions in the symbiosis of cnidarians. (A) Summary of symbiotic interactions between *Hydra* and *Chlorella* A99. During light conditions, *Chlorella* A99 performs photosynthesis and produces maltose (Mal), which is secreted into the *Hydra* symbiosome where it is possibly digested to glucose (Gluc), shown in red. The sugar induces expression of *Hydra* genes encoding glutamine synthetase (GS), Na/Pi transporter (NaPi) and Spot14. GS catalyzes the condensation of glutamate (Glu) and ammonium ($NH_4^+$) to form glutamine (Gln), which is used by *Chlorella* as a nitrogen source. Since the sugar also up-regulates the NaPi gene, which controls intracellular phosphate levels, it might be involved in the supply of phosphorus to *Chlorella* as well (blue broken line). The sugar is transported to the ectoderm (red broken line) and there induces the expression of GS and Spot14. In the *Chlorella* A99 genome, degeneration of the nitrate assimilation system and an increase of amino acid transporters was observed (green balloon). (B, C) Comparison between *Hydra-Chlorella* symbiosis and coral-*Symbiodinium* symbiosis. Red indicates transfer of photosynthesis products from the symbiont to the host, and blue indicates transfer of nitrogen sources from the host to the symbiont. While the host organisms *Hydra* and coral can assimilate $NH_4^+$ to Gln (B, C), assimilation of inorganic nitrogen by *Symbiodinidium* plays an important role for the symbiotic system in coral (C).
DOI: https://doi.org/10.7554/eLife.35122.032

the nutrients provided by the symbiont to the host. For example, symbiotic *Chlorella* algae in green hydra, *Paramecium* and fresh water sponges provide their photosynthetic products in form of maltose and glucose (*Figure 8B*) (*Brown and Nielsen, 1974*; *Wilkinson, 1980*; *Kamako and Imamura, 2006*). In contrast, *Symbiodinium* provides glucose, glycerol, organic acids, amino acids as well as lipids to its marine hosts (*Figure 8C*) (*Muscatine and Cernichiari, 1969*; *Lewis and Smith, 1971*; *Trench, 1971*; *Kellogg and Patton, 1983*). A former transcriptome analysis of amino acid biosynthetic pathways suggested that *Symbiodinium* can synthesize almost all amino acids (*Shinzato et al., 2014*). Gene loss in cysteine synthesis pathway in the coral host *Acropora digitifera* seems to reflect the dependency on the amino acids provided by the *Symbiodinium* symbiont (*Shinzato et al., 2011*). In contrast to *Symbiodinium* which can assimilate inorganic nitrogen such as nitrate and

ammonium (*Lipschultz and Cook, 2002*; *Grover et al., 2003*; *Tanaka et al., 2006*; *Yellowlees et al., 2008*), the symbiotic *Chlorella* algae in *Hydra* and *Paramecium* can only use amino acids as a nitrogen source (*Figure 6*) (*Kamako et al., 2005*).

In efforts to explain the metabolic efficiency of nitrogen use in symbiotic organisms, two models have been proposed: the 'nitrogen conservation' and the 'nitrogen recycling' hypothesis. The nitrogen conservation hypothesis suggests that photosynthetic carbon compounds from the symbiont are used preferentially by the host respiration, which reduces catabolism of nitrogenous compounds (*Rees and Ellard, 1989*; *Szmant et al., 1990*; *Wang and Douglas, 1998*). The 'nitrogen recycling' hypothesis suggests that symbionts assimilate nitrogenous waste (ammonium) of the host into valuable, organic compounds, which then are translocated back to the host (*Figure 8C* Symbiont nitrogen assimilation) (*Lewis and Smith, 1971*; *Muscatine and Porter, 1977*; *Falkowski et al., 1993*; *Wang and Douglas, 1998*). Our observation that in symbiotic green hydra many genes involved in amino acid metabolism are down-regulated (*Figure 1E*) is consistent with the assumption of reduction of amino acid consumption by respiration.

In addition to the nitrogen recycling hypothesis, it has been proposed that also corals, sea anemones, *Paramecium* and green hydra hosts can assimilate ammonium into amino acids (*Figure 8B,C* Host nitrogen assimilation) (*Miller and Yellowlees, 1989*; *Rees, 1989*; *Szmant et al., 1990*; *Rees, 1991*; *Wang and Douglas, 1998*; *Lipschultz and Cook, 2002*). Ammonia assimilation by the host implies that the host controls the nitrogen status to regulate metabolism of the symbionts, which may be involved in controlling the number of symbionts within the host cell. For organisms such as corals living in oligotrophic sea, inorganic nitrogen assimilation and recycling may be necessary to manage the nitrogen sources efficiently. In contrast, for *Hydra* and *Paramecium* living in a relatively nutrient-rich environment may be advantageous in terms of metabolic efficiency that the symbiont abandons its ability to assimilate inorganic nitrogen and specializes in the supply of photosynthetic carbohydrate to the host.

## Genome evolution in symbiotic *Chlorella* sp. A99

Metabolic dependence of symbionts on host supply occasionally results in genome reduction and gene loss. For example, symbiotic *Buchnera* bacteria in insects are missing particular genes in essential amino acid pathways (*Shigenobu et al., 2000*; *Hansen and Moran, 2011*). The fact that the corresponding genes of the host are up-regulated in the bacteriocyte, indicates complementarity and syntrophy between host and symbiont. Similarly, in *Chlorella* A99 the nitrogen assimilation system could have been lost as a result of continuous supply of nitrogenous amino acids provided by *Hydra*.

Compared to *Chlorella* NC64A, the closest relative to *Chlorella* A99 among the genome-sequenced algae, genome size and total number of genes in *Chlorella* A99 were found to be smaller (*Figure 4B*). Although both A99 and NC64A cannot be cultivated using inorganic nitrogen sources (*Figure 7*) (*Kamako et al., 2005*), NC64A, unlike A99, obtains all major nitrogen assimilation genes and their cluster structure on the chromosome (*Figure 6*) (*Sanz-Luque et al., 2015*). NR and NiR activities were found to be induced by nitrate in free-living *Chlorella*, but not in *Chlorella* NC64A, indicating mutations in the regulatory region (*Kamako et al., 2005*). Considering the phylogenetic position of NC64A and the symbiotic *Chlorella* of green hydra (*Kawaida et al., 2013*), the disability of nitrate assimilation in A99 and NC64A seems to have evolved independently, suggesting convergent evolution in a similar symbiotic environment.

Although our findings indicate that genome reduction in *Chlorella* A99 is more advanced than in *Chlorella* NC64A, genome size and total number of genes do not differ much between the Trebouxiophyceae (A99, NC64A and C169) (*Figure 4B*). By contrast, the parasitic algae *Helicosporidium* and *Auxochlorella* have significantly smaller genome sizes and number of genes indicating extensive genome reduction (*Gao et al., 2014*; *Pombert et al., 2014*). The apparently unchanged complexity of the *Chlorella* A99 genome suggests a relatively early stage of this symbiotic partnership. Thus, gene loss in metabolic pathways could occur as a first step of genome reduction in symbionts caused by the adaptation to continuous nutrient supply from the host. Taken together, our study suggests metabolic-codependency as the primary driving force in the evolution of symbiosis between *Hydra* and *Chlorella*.

# Materials and methods

## Key resources table

| Reagent type (species) or resource | Designation | Source or reference | Identifiers | Additional information |
|---|---|---|---|---|
| Strain, strain background (*Hydra viridissima* A99) | *Hydra viridissima* A99 | PMID: 16351895 | | |
| Strain, strain background (*Chlorella* sp. A99) | *Chlorella* sp. A99 | PMID: 16351895 | NCBI BioProject ID: PRJNA412448 | |
| Strain, strain background (*Chlorella variabilis* NC64A) | *Chlorella variabilis* NC64A | Microbial Culture Collection at the National Institute for Environmental Studies | NIES-2541 | |
| Strain, strain background (*Coccomyxa subellipsoidea* C-169) | *Coccomyxa subellipsoidea* C-169 | Microbial Culture Collection at the National Institute for Environmental Studies | NIES-2166 | |
| Strain, strain background (*Chlamydomonas reinhardtii*) | *Chlamydomonas reinhardtii* | Microbial Culture Collection at the National Institute for Environmental Studies | NIES-2235 | |
| Commercial assay or kit | TruSeq DNA LT Sample Prep Kit | Illumina | FC-121–2001 | |
| Commercial assay or kit | Nextera Mate Pair Sample Preparation Kit | Illumina | FC-132–1001 | |
| Commercial assay or kit | Miseq reagent kit v3 | Illumina | MS-102–3003 | |
| Commercial assay or kit | HiSeq SBS kit v4 | Illumina | FC-401–4003 | |
| Commercial assay or kit | BigDye Terminator v3.1 Cycle Sequencing Kit | Thermo Fisher Scientific | 4337454 | |
| Commercial assay or kit | 4 × 44K *Hydra viridissima* A99 Custom-Made Microarray | Agilent Technologies | NCBI GEO Platform ID: GPL23280 | |
| Commercial assay or kit | GE Hybridization Kit and GE Wash Pack | Agilent Technologies | 5188–5242, 5188–5327 | |
| Commercial assay or kit | High Sensitivity DNA Kit | Agilent Technologies | 5067–4626 | |
| Commercial assay or kit | RNA6000 nano kit | Agilent Technologies | 5067–1511 | |
| Commercial assay or kit | Low Input Quick Amp Labeling Kit | Agilent Technologies | 5190–2305 | |
| Commercial assay or kit | PureLink RNA Mini Kit | Thermo Fisher Scientific | 12183018A | |
| Commercial assay or kit | Fermentas First Strand cDNA Synthesis Kit | Thermo Fisher Scientific | K1621 | |
| Chemical compound, drug | Trizol reagent | Thermo Fisher Scientific | 15596026 | |
| Chemical compound, drug | AmpliTaq Gold 360 Master Mix | Thermo Fisher Scientific | 4398901 | |
| Chemical compound, drug | ISOPLANT II | Nippon Gene | 316–04153 | |
| Chemical compound, drug | GoTaq qPCR Master Mix | Promega | A6002 | |
| Chemical compound, drug | KOD FX Neo | TOYOBO | KFX-201 | |
| Software, algorithm | Feature Extraction Software | Agilent Technologies | RRID:SCR_014963 | |
| Software, algorithm | Newbler | 454 Life Sciences, Roche Diagnostics | RRID:SCR_011916 | |
| Software, algorithm | SSPACE | PMID: 21149342 | RRID:SCR_005056 | |
| Software, algorithm | GapCloser | PMID: 23587118 | RRID:SCR_015026 | |
| Software, algorithm | NCBI BLAST | PMID: 2231712 | RRID:SCR_004870 | |
| Software, algorithm | CEGMA | PMID: 17332020 | RRID:SCR_015055 | |

*Continued on next page*

*Continued*

| Reagent type (species) or resource | Designation | Source or reference | Identifiers | Additional information |
|---|---|---|---|---|
| Software, algorithm | Augustus: Gene Prediction | PMID: 16845043 | RRID:SCR_008417 | |
| Software, algorithm | Blast2GO | PMID: 16081474 | RRID:SCR_005828 | |
| Software, algorithm | Hmmer | PMID: 9918945 | RRID:SCR_005305 | |
| Software, algorithm | CLUSTALX2 | PMID: 17846036 | RRID:SCR_002909 | |
| Software, algorithm | BioEdit | Nucleic Acid Symposium Series 41, 95–98 | RRID:SCR_007361 | |
| Software, algorithm | Njplot | Biochimie 78, 364–369 | NA | |
| Software, algorithm | OrthoFinder | PMID: 26243257 | NA | |

## Biological materials and procedures

Experiments were carried out with the Australian *Hydra viridissima* strain A99, which was obtained from Dr. Richard Campbell, Irvine. Polyps were maintained at 18°C on a 12 hr light/dark cycle and fed with *Artemia* two or three times a week. Aposymbiotic (algae free) polyps were obtained by photobleaching using 5 µM DCMU (3-(3,4-dichlorophenyl)−1,1-dimethylurea) as described before (*Pardy, 1976*; *Habetha et al., 2003*). Experiments were carried out with polyps starved for 3–6 days. Isolation of endodermal layer and ectodermal layer was performed as described by Kishimoto et al. (*Kishimoto et al., 1996*). Symbiotic *Chlorella* were isolated as described before by Muscatine and McAuley (*Muscatine, 1983*; *McAuley, 1986*). *Chlorella variabilis* NC64A (NIES-2541), *Coccomyxa subellipsoidea* C-169 (NIES-2166) and *Chlamydomonas reinhardtii* (NIES-2235) were obtained from the Microbial Culture Collection at the National Institute for Environmental Studies (Tsukuba, Japan).

## Nucleic acid preparation

Total RNA of *Hydra* was extracted by use of the Trizol reagent and PureLink RNA Mini Kit (Thermo Fisher Scientific) after lysis and removal of algae by centrifugation. The genomic DNA of green algae was extracted using ISOPLANT II (Nippon Gene, Tokyo, Japan) following DNase I treatment to degrade contaminant DNA. Quantity and quality of DNA and RNA were checked by NanoDrop (Thermo Scientific Inc., Madison, USA) and BioAnalyzer (Agilent Technologies, Santa Clara, USA).

## Microarray analysis

Total RNA for synthesis of cRNA targets was extracted from about 100 green hydra for each experimental group. Experiments were carried out using three biological replicates. cRNA labeled with cyanine-3 were synthesized from 400 ng total *Hydra* RNA using a Low Input Quick Amp Labeling Kit for one color detection (Agilent Technologies). A set of fluorescently labeled cRNA targets was employed in a hybridization reaction with 4 × 44K Custom-Made *Hydra viridissima* Microarray (Agilent Technologies) contributing a total of 43,222 transcripts that was built by mRNA-seq data (NCBI GEO Platform ID: GPL23280) (*Bosch et al., 2009*). Hybridization and washing were performed using the GE Hybridization Kit and GE Wash Pack (Agilent Technologies) after which the arrays were scanned on an Agilent Technologies G2565BA microarray scanner system with SureScan technology following protocols according to the manufacturer's instructions. The intensity of probes was extracted from scanned microarray images using Feature Extraction 10.7 software (Agilent Technologies). All algorithms and parameters used in this analysis were used with default conditions. Background-subtracted signal-intensity values (gProcessedSignal) generated by the Feature Extraction software were normalized using the 75$^{th}$ percentile signal intensity among the microarray. Those genes differentially expressed between two samples were determined by average of fold change (cut of >2.0) and Student's t-test ($p < 0.1$). The data series are accessible at NCBI GEO under accession number GSE97633.

## Quantitative real time RT-PCR

Total RNA was extracted from 50 green hydra polyps for each biological replicate independently. For reverse transcription of total RNA First Strand cDNA Synthesis Kit (Fermentas, Ontario, Canada)

was used. Real-time PCR was performed using GoTaq qPCR Master Mix (Promega, Madison, USA) and ABI Prism 7300 (Applied Biosystems, Foster City, USA). All qPCR experiments were performed in duplicate with three biological replicates each. Values were normalized using the expression of the tubulin alpha gene. Primers used for these experiments are listed in *Supplementary file 6A*.

## Whole mount in situ hybridization

Expression patterns of specific *Hydra* genes were detected by whole mount in situ hybridization with digoxigenin (DIG)-labelled RNA probes. Specimens were fixed in 4% paraformaldehyde. Hybridization signal was visualized using anti-DIG antibodies conjugated to alkaline phosphatase and NBT/BCIP staining solution (Roche). DIG-labeled sense probes (targeting the same sequences as the anti-sense probes) were used as a control. Primers used for these experiments are listed in *Supplementary file 6B*.

## Genome sequencing and gene prediction

For genome sequencing of *Chlorella* sp. A99, *Chlorella* sp. A99 was isolated from *H. viridissima* A99 and genomic DNA was extracted. Paired-end library (insert size: 740 bp) and mate-pair libraries (insert size: 2.2 and 15.2 kb) were made using Illumina TruSeq DNA LT Sample Prep Kit and Nextera Mate Pair Sample Preparation Kit respectively (Illumina Inc., San Diego, USA), following the manufacturer's protocols. Genome sequencing was performed using Illumina Miseq and Hiseq 2000 platforms. Sequence reads were assembled using Newbler Assembler version 2.8 (Roche, Penzberg, Germany) and subsequent scaffolding was performed by SSPACE (*Boetzer et al., 2011*). Gaps inside the scaffolds were closed with the paired-end and mate-pair data using GapCloser of Short Oligonucleotide Analysis Package (*Luo et al., 2012*). To overcome potential assembly errors arising from tandem repeats, sequences that aligned to another sequence by more than 50% of the length using blastn (1e-50) were removed from the assembly. The completeness of the genome was evaluated using CEGMA v2.4 (Core Eukaryotic Genes Mapping Approach) based on mapping of the 248 most highly conserved core eukaryotic genes (CEGs) on the assembled genome (*Parra et al., 2007*). The completeness of complete and partial CEGs in the A99 scaffolds was 80 and 88%, respectively. The fraction of repetitive sequences was 12%. Gene model was predicted by AUGUSTUS 3.0.1 using model parameters for NC64A (*Stanke et al., 2006*). This Whole Genome Shotgun project has been deposited at DDBJ/ENA/GenBank under the accession PCFQ00000000 (BioProject ID: PRJNA412448). Genome sequences and gene models are also accessible at the website of OIST Marine Genomics Unit Genome Project (http://marinegenomics.oist.jp/chlorellaA99/viewer/info?project_id=65).

## Analysis of genes in *Hydra viridissima* and *Chlorella*

Annotation of transcriptome contigs and prediction of gene models was performed by use of BLAST, Gene Ontology (*Ashburner et al., 2000*) and blast2go (*Conesa et al., 2005*). To examine the conservation of *H. viridissima* contigs among metazoans, homology searches by blastx (evalue 1E-5) were performed using protein databases obtained from NCBI for *Drosophila melanogaster* and *Homo sapiens*, from the JGI genome portal (http://genome.jgi.doe.gov/) for *Branchiostoma floridae*, *Nematostella vectensis*, from Echinobase (http://www.echinobase.org/EchinoBase/) for *Strongylocentrotus pupuratus*, from Compagen for *Hydra magnipapillata*, and from the OIST marine genomics Genome browser ver.1.1 (http://marinegenomics.oist.jp/coral/viewer/info?project_id=3) for *Acropora digitifera*.

For comparative analysis of gene models of *Chlorella* sp. A99 and other algae, domain searches against the Pfam database (Pfam-A.hmm) were performed using HMMER (*Eddy, 1998*; *Finn et al., 2016*), and ortholog gene grouping was done using OrthoFinder (*Emms and Kelly, 2015*). The sequences of the reference genes and genomes were obtained from the database of the JGI genome portal for *Chlorella variabilis* NC64A, *Coccomyxa subellipsoidea* C-169, *Volvox carteri*, *Micromonas pusilla*, and *Ostreococcus tauri*, from NCBI for *Auxenochlorella protothecoides* 0710, from Phytozome (http://phytozome.jgi.doe.gov/pz/portal.html) for *Chlamydomonas reinhardtii*, from OIST Marine Genomics (http://marinegenomics.oist.jp/symb/viewer/info?project_id=21) for *Symbiodinium minutum*, *Symbiodinium kawagutti* genome, from Dinoflagellate Resources (http://web.malab.cn/symka_new/) for *Symbiodinium kawagutti* and Reefgenomics (http://reefgenomics.org/)

for *Symbiodinium microadriaticum*) (*Merchant et al., 2007*; *Palenik et al., 2007*; *Worden et al., 2009*; *Blanc et al., 2010*; *Prochnik et al., 2010*; *Blanc et al., 2012*)

Nitrogen assimilation genes in *Chlorella* A99 were identified by orthologous gene groups and reciprocal blast searches. The number of genes for nitrate assimilation genes, glutamine synthetase and glutamate synthetase, and clustering of such genes were systematically reported by (*Sanz-Luque et al., 2015*). We used these data as reference for searches of nitrogen assimilation genes, and further nitrogen assimilation genes were searched by Kyoto Encyclopedia of Genes and Genomes (KEGG) pathway (*Kanehisa and Goto, 2000*). JGI genome browsers of *Chlorella variabilis* NC64A and *Coccomyxa subellipsoidea* C-169 were also used for retrieving genes and checking gene order on the scaffolds.

## Phylogenetic analysis

For a phylogenetic tree of chlorophyte green algae, the sequences of 18S rRNA gene, ITS1, 5.8S rRNA gene, ITS2 and 28S rRNA gene were obtained from scaffold20 of *Chlorella* A99 genome sequence, and from NCBI nucleotide database entries for *Chlorella variabilis* NC64A (FM205849.1), *Auxenochlorella prototothecoides* 0710 (NW_011934479.1), *Coccomyxa subellipsoidea* C169 (AGSI01000011.1), *Volvox carteri* f. nagariensis (NW_003307662.1), *Chlamydomonas reinhardtii* (FR865576.1), *Ostreococcus tauri* (GQ426340.1) and *Micromonas pusilla* (FN562452.1). Multiple alignments were produced with CLUSTALX (2.1) with gap trimming (*Larkin et al., 2007*). Sequences of poor quality that did not well align were deleted using BioEdit (*Hall, 1999*). Phylogenetic analyses were performed using the Neighbor-Joining method by CLUSTALX with the default parameters (1000 bootstrap tests and 111 seeds). Representative phylogenetic trees were drawn by using NJ plot (*Perrière and Gouy, 1996*).

## PCR amplification of nitrate assimilation genes in green algae

Primers were designed based on the conserved region of the NRT2 gene, NiR and NR genes (positive control) identified by comparison of genes from *Chlorella variabilis* NC64A (NC64A), *Coccomyxa subellipsoidea* C169 (C169), and *Chlamydomonas reinhardtii* (Cr) which belongs to Chlorophyceae class of green algae. Primers for NAR2 could not be designed because of insufficient conservation. As positive controls, amplicons were produced for NR of all the green algae examined and of NRT2 and NiR from NC64A, C169 and Cr, after which their sequences were checked. KOD FX Neo (TOYOBO, Tokyo, Japan) was used under the following conditions: an initial denaturation phase (94°C for 120 s) followed by 36 cycles of (98°C for 30 s, 69°C for 100 s) for NiR, (98°C for 30 s, 58°C for 30 s and 68°C for 210 s) for NRT2 and (98°C for 30 s, 59°C for 30 s and 68°C for 60 s) for NR. In each case, 10 ng gDNA was used as a template. The primers used are described in *Supplementary file 6C*. PCR products were sequenced to confirm amplification of the target genes using ABI PRISM 3100 Genetic Analyzer (Thermo Fisher Scientific Inc., Madison, USA) using BigDye Terminator v3.1 Cycle Sequencing Kit (Thermo Fisher Scientific).

## In vitro culture of algae

To isolate symbiotic algae, polyps were quickly homogenized in 0.25% sodium dodecyl sulfate (SDS) solution and centrifuged at 3000 g for 1 min. The pellet was resuspended in 0.05% SDS and centrifuged at 500 g for 5 min. Isolated A99, NC64A and C169 were washed by sterilized Bold Basal Medium (*Bischoff and Bold, 1963*) modified by the addition of 0.5% glucose, 1.2 mg/L vitamine B1 (Thiaminhydrochloride), 0.01 mg/L vitamine B12 (Cyanocobalamin) (*Supplementary file 7*) and incubated for two days in modified Bold Basal Medium with 50 mg/l ampicillin and streptomycin. The algae were cultivated in 5 ml of modified Bold Basal Medium (BBM) with the same amount of nitrogen (2.9 mM NaNO$_3$, NH$_4$Cl, glutamine or 426 mg/l casamino acids) and 5 mg/l Carbendazim (antifungal) with fluorescent illumination (12 hr light, 12 hr dark) at 20°C. Mean numbers of algae per ml were calculated from three tubes enumerated at 4, 8, and 12 days after inoculation with 10$^6$ cell/sml using a hemocytometer. After cultivation, gDNA was isolated from the A99 cultured in Gln-containing BBM and casamino acid-containing BBM and A99 was isolated from green hydra directly. A partial genomic region of the 18S rRNA gene was amplified by PCR and sequenced to confirm absence of contamination by other algae. PCR was performed using AmpliTaq Gold (Thermo Fisher

Scientific). Sequencing was performed as described above. The primers used are described in **Supplementary file 6D**.

## Acknowledgements

We thank Trudy Wassenaar for critical reading the text and for discussion. We also thank the DNA Sequencing Section, IT Section and Kanako Hisata in the Okinawa Institute of Science and Technology (OIST) for excellent technical support. The computations for this work were partially performed on the NIG supercomputer at the ROIS National Institute of Genetics. We are thankful to Angela Douglas for sustained exchanges and discussions on symbiosis in hydra and two anonymous referees for their constructive comments on a previous version of this manuscript. This work was supported in part by JSPS KAKENHI Grant-in-Aid for Young Scientists (B) 25840132 and Scientific Research (C) 15K07173 to MH. and by the Deutsche Forschungsgemeinschaft (DFG) (CRC1182 'Origin and Function of Metaorganisms'). TCGB. gratefully appreciates support from the Canadian Institute for Advanced Research (CIFAR).

## Additional information

### Funding

| Funder | Grant reference number | Author |
| --- | --- | --- |
| Japan Society for the Promotion of Science | Young Scientists (B) 25840132 | Mayuko Hamada |
| Japan Society for the Promotion of Science | Scientific Research (C) 15K07173 | Mayuko Hamada |
| Deutsche Forschungsgemeinschaft | CRC1182 | Thomas CG Bosch |

The funders had no role in study design, data collection and interpretation, or the decision to submit the work for publication.

### Author contributions

Mayuko Hamada, Conceptualization, Formal analysis, Funding acquisition, Investigation, Visualization, Writing—original draft; Katja Schröder, Formal analysis, Investigation, Visualization, Writing—review and editing; Jay Bathia, Formal analysis, Investigation, Writing—review and editing; Ulrich Kürn, Formal analysis, Investigation; Sebastian Fraune, Resources, Writing—review and editing; Mariia Khalturina, Investigation, Writing—review and editing, Library preparation and genome sequencing; Konstantin Khalturin, Chuya Shinzato, Formal analysis, Writing—review and editing; Nori Satoh, Supervision, Project administration, Writing—review and editing; Thomas CG Bosch, Conceptualization, Supervision, Funding acquisition, Writing—original draft, Writing—review and editing

### Author ORCIDs

Mayuko Hamada http://orcid.org/0000-0001-7306-2032
Katja Schröder http://orcid.org/0000-0003-1158-2598
Sebastian Fraune http://orcid.org/0000-0002-6940-9571
Konstantin Khalturin http://orcid.org/0000-0003-4359-2993
Chuya Shinzato http://orcid.org/0000-0001-7843-3381
Nori Satoh https://orcid.org/0000-0002-4480-3572
Thomas CG Bosch http://orcid.org/0000-0002-9488-5545

### Decision letter and Author response

Decision letter https://doi.org/10.7554/eLife.35122.064
Author response https://doi.org/10.7554/eLife.35122.065

# Additional files

## Supplementary files

• Supplementary file 1. Results of microarray analysis and list of differentially expressed genes. Gene expression of green hydra with native symbiotic *Chlorella* A99 (Hv_Sym), that in sexual phase (Hv_Sym_sexy), aposymbiotic polyps from which symbiotic *Chlorella* were removed (Hv_Apo) and aposymbiotic polyps reinfected with *Chlorella variabilis* NC64A (Hv_NC64A) were compared.
DOI: https://doi.org/10.7554/eLife.35122.033

• Supplementary file 2. (A) Ortholog groups of Aa_trans containing protein in *Chlorella variabilis NC64A* (NC64A), *Coccomyxa subellipsoidea* C-169 (C169), *Chlamydomonas reinhardtii* (Cr), *Volvox carteri* (Vc), *Micromonas pusilla* (Mp) and *Ostreococcus tauri* (Ot). (B) Blast best hit genes of *Arabidopsis thaliana* in *Chlorella* sp. A99 genes belonging to OG0000040 and OG0000324.
DOI: https://doi.org/10.7554/eLife.35122.034

• Supplementary file 3. List of *Coccomyxa subellipsoidea* C169 (C169) and their BLAST best hit genes in *Chlamydomonas reinhardtii* (Cr), *Volvox carteri* (Vc) and *Chlorella* A99 (A99) gene model and genome scaffolds.
DOI: https://doi.org/10.7554/eLife.35122.035

• Supplementary file 4. Sequence ID of nitrogen assimilation genes in *Symbiodinium*.
DOI: https://doi.org/10.7554/eLife.35122.036

• Supplementary file 5. Sequence ID of nitrogen assimilation genes in *Chlorella variabilis NC64A* (NC64A), *Coccomyxa subellipsoidea* C-169 (C169), *Volvox carteri* (Vc), *Micromonas pusilla* (Mp) and *Ostreococcus tauri* (Ot) and *Chlamydomonas reinhardtii* (Cr).
DOI: https://doi.org/10.7554/eLife.35122.037

• Supplementary file 6. Primers used in this study, for quantitative real time RT-PCR. (A), in situ hybridization probes (B), PCR amplification of nitrogen assimilation genes in green algae (C) and PCR amplification of 18S ribosomal DNA gene in green algae (D).
DOI: https://doi.org/10.7554/eLife.35122.038

• Supplementary file 7. Composition of modified Bold's Basal Medium for one liter (pH. 7).
DOI: https://doi.org/10.7554/eLife.35122.039

• Transparent reporting form
DOI: https://doi.org/10.7554/eLife.35122.040

## Data availability

Microarray information and the data series are accessible at NCBI GEO under accession number GPL23280 and GSE97633 respectively. All the results of microarray analysis are included in Supplementary Table 1. The Whole Genome Shotgun project of Chlorella sp. A99 has been deposited at DDBJ/ENA/GenBank under the accession PCFQ00000000 (BioProject ID: PRJNA412448). Genome sequences and gene models are also accessible at the website of OIST Marine Genomics Unit Genome Project (http://marinegenomics.oist.jp/chlorellaA99/viewer/info?project_id=65). All data generated by qPCR are included in Source Data: Figure2, Figure2 - Figure supplement 1, Source Data: Figure3, Source Data: Figure3 - Figure Supplement 2 and Source Data: Table 2, Table 4

The following datasets were generated:

| Author(s) | Year | Dataset title | Dataset URL | Database, license, and accessibility information |
|---|---|---|---|---|
| Mayuko Hamada | 2018 | Chlorella sp. A99 genome sequence and gene models | http://marinegenomics.oist.jp/chlorellaA99/viewer/info?project_id=65 | Publicly available at OIST Marine Genomics Unit (Chlorella sp. A99) |
| Mayuko Hamada | 2018 | Chlorella sp. A99 genome sequence | http://www.ncbi.nlm.nih.gov/bioproject/412448 | Publicly available at NCBI BioProject (Accession no: PCFQ00000000) |
| Fraune S, Bosch TC | 2017 | Agilent-029560 Hydra viridissima transcriptome-based custom microarray | https://www.ncbi.nlm.nih.gov/geo/query/acc.cgi?acc=GPL23280 | Publicly available at the NCBI Gene Expression Omnibus |

| | | | | (accession no: GPL23280) |
|---|---|---|---|---|
| Fraune S, Kürn U, Bosch TC | 2017 | Identification of genes involved in symbiosis of green hydra and Chlorella | https://www.ncbi.nlm.nih.gov/geo/query/acc.cgi?acc=GSE97633 | Publicly available at the NCBI Gene Expression Omnibus (accession no: GSE97633) |

The following previously published datasets were used:

| Author(s) | Year | Dataset title | Dataset URL | Database, license, and accessibility information |
|---|---|---|---|---|
| Blanc G, Duncan G, Agarkova I, Borodovsky M, Gurnon J, Kuo A, Lindquist E, Lucas S, Pangilinan J, Polle J, Salamov A, Terry A, Yamada T, Dunigan DD, Grigoriev IV, Claverie JM, Van Etten JL | 2010 | The Chlorella variabilis NC64A genome reveals adaptation to photosymbiosis, coevolution with viruses, and cryptic sex | https://genome.jgi.doe.gov/ChlNC64A_1/ChlNC64A_1.home.html | Publicly available at JGI MycoCosm (Chlorella variabilis NC64A) |
| Blanc G, Agarkova I, Grimwood J, Kuo A, Brueggeman A, Dunigan DD, Gurnon J, Ladunga I, Lindquist E, Lucas S, Pangilinan J, Proschold T, Salamov A, Schmutz J, Weeks D, Yamada T, Lomsadze A, Borodovsky M, Claverie JM, Grigoriev IV, Van Etten JL | 2012 | The genome of the polar eukaryotic microalga Coccomyxa subellipsoidea reveals traits of cold adaptation | https://genome.jgi.doe.gov/Coc_C169_1/Coc_C169_1.home.html | Publicly available at JGI MycoCosm (Coccomyxa subellipsoidea C-169 v2.0) |
| Prochnik SE, Umen J, Nedelcu AM, Hallmann A, Miller SM, Nishii I, Ferris P, Kuo A, Mitros T, Fritz-Laylin LK, Hellsten U, Chapman J, Simakov O, Rensing SA, Terry A, Pangilinan J, Kapitonov V, Jurka J, Salamov A, Shapiro H, Schmutz J, Grimwood J, Lindquist E, Lucas S, Grigoriev IV, Schmitt R, Kirk D, Rokhsar DS | 2010 | Genomic analysis of organismal complexity in the multicellular green alga Volvox carteri | https://phytozome.jgi.doe.gov/pz/portal.html#!info?alias=Org_Vcarteri | Publicly available at JGI Phytozome (Volvox carteri v2.1) |
| Merchant SS, Prochnik SE, Vallon O, Harris EH, Karpowicz SJ, Witman, GB, Terry A, Salamov, A, Fritz-Laylin LK, Marechal-Drouard L, Marshall WF, Qu LH, Nelson DR, Sanderfoot AA, Spalding MH, Kapitonov VV, Ren Q, Ferris P, Lindquist | 2007 | The Chlamydomonas genome reveals the evolution of key animal and plant functions | http://phytozome.jgi.doe.gov/pz/portal.html | Publicly available at JGI Phytozome (Chlamydomonas reinhardtii v5.5) |

| | | | | |
|---|---|---|---|---|
| E, Shapiro H, Lucas SM, Grimwood J, Schmutz J, Cardol P, Cerutti H, Chanfreau G, Chen CL, Cognat V, Croft MT, Dent R, Dutcher S, Fernandez E, Fukuzawa H, Gonzalez-Ballester D, Gonzalez-Halphen D, Hallmann A, Hanikenne M, Hippler M, Inwood W, Jabbari K, Kalanon M, Kuras R, Lefebvre PA, Lemaire SD, Lobanov AV, Lohr M, Manuell A, Meier I, Mets L, Mittag M, Mittelmeier T, Moroney JV, Moseley J, Napoli C, Nedelcu AM, Niyogi K, Novoselov SV, Paulsen IT, Pazour G, Purton S, Ral JP, Riano-Pachon DM, Riekhof W, Rymarquis L, Schroda M, Stern D, Umen J, Willows R, Wilson N, Zimmer SL, Allmer J, Balk J, Bisova K, Chen CJ, Elias M, Gendler K, Hauser C, Lamb MR, Ledford H, Long JC, Minagawa J, Page MD, Pan J, Pootakham W, Roje S, Rose A, Stahlberg E, Terauchi AM, Yang P, Ball S, Bowler C, Dieckmann CL, Gladyshev VN, Green P, Jorgensen R, Mayfield S, Mueller-Roeber B, Rajamani S, Sayre RT, Brokstein P, Dubchak I, Goodstein D, Hornick L, Huang YW, Jhaveri J, Luo Y, Martinez D, Ngau WC, Otillar B, Poliakov A, Porter A, Szajkowski L, Werner G, Zhou, K, Grigoriev IV, Rokhsar DS, Grossman AR | | | | |
| Worden AZ, Lee JH, Mock T, Rouze P, Simmons MP, Aerts AL, Allen AE, Cuvelier ML, Derelle E, Everett MV, Foulon E, Grimwood J, Gundlach H, Henrissat B, | 2009 | Green evolution and dynamic adaptations revealed by genomes of the marine picoeukaryotes Micromonas | https://genome.jgi.doe.gov/MicpuC2/MicpuC2.home.html | Publicly available at JGI MycoCosm (Micromonas pusilla CCMP1545) |

| | | | | |
|---|---|---|---|---|
| Napoli C, McDonald SM, Parker MS, Rombauts S, Salamov A, Von Dassow P, Badger JH, Coutinho PM, Demir E, Dubchak I, Gentemann C, Eikrem W, Gready JE, John U, Lanier W, Lindquist EA, Lucas S, Mayer KF, Moreau H, Not F, Otillar R, Panaud O, Pangilinan J, Paulsen I, Piegu B, Poliakov A, Robbens S, Schmutz J, Toulza E, Wyss T, Zelensky A, Zhou K, Armbrust EV, Bhattacharya D, Goodenough UW, Van de Peer Y, Grigoriev IV | | | | |
| Palenik B, Grimwood J, Aerts A, Rouze P, Salamov A, Putnam N, Dupont C, Jorgensen R, Derelle E, Rombauts S, Zhou K, Otillar R, Merchant SS, Podell S, Gaasterland T, Napoli C, Gendler K, Manuell A, Tai V, Vallon O, Piganeau G, Jancek S, Heijde M, Jabbari K, Bowler C, Lohr M, Robbens S, Werner G, Dubchak I, Pazour GJ, Ren Q, Paulsen I, Delwiche C, Schmutz J, Rokhsar D, Van de Peer Y, Moreau H, Grigoriev IV | 2007 | The tiny eukaryote Ostreococcus provides genomic insights into the paradox of plankton speciation | https://genome.jgi.doe.gov/Ostta4221_3/Ostta4221_3.home.html | Publicly available at JGI MycoCosm (Ostreococcus tauri RCC4221 v3.0) |
| Finn RD, Coggill P, Eberhardt RY, Eddy SR, Mistry J, Mitchell AL, Potter SC, Punta M, Qureshi M, Sangrador-Vegas A, Salazar GA, Tate J, Bateman A | 2016 | The Pfam protein families database: towards a more sustainable future. Nucleic Acids Res. 44, D279-285. 10.1093/nar/gkv1344. | ftp://ftp.ebi.ac.uk/pub/databases/Pfam/releases/Pfam31.0/ | Available to download at the ftp site. (dataset name: Pfam-A.hmm) |

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
