## [Decision Letter]

Thank you for submitting your article "Metabolic co-dependence drives the evolutionary ancient *Hydra-Chlorella* symbiosis" for consideration by *eLife*. Your article has been reviewed by two peer reviewers, and the evaluation has been overseen by a Reviewing Editor and Ian Baldwin as the Senior Editor. The following individuals involved in review of your submission have agreed to reveal their identity: Virginia M Weis (Reviewer #1); Eunsoo Kim (Reviewer #2).

The reviewers have discussed the reviews with one another and the Reviewing Editor has drafted this decision to help you prepare a revised submission.

Summary:

This manuscript describes genomic and transcriptomic evidence for the metabolic interdependence of the partners in the *Hydra-Chlorella* symbiosis. The work was carefully designed and executed and it is presented in a well-constructed and well-written manuscript. The genome based investigation confirmed that the consortium is driven by nutrient exchanges: some animal genes identified in this study have implications in carbohydrate metabolism whereas the algal genome suggests its reduced competency in assimilation of inorganic source of nitrogen and an increased capacity to transport amino acids. The work produced, among other things, a high-quality genome assembly for the symbiotic *Chlorella* sp.

The study casts the work in an evolutionary and historical light. The researchers do a good job of capturing the excellent original green Hydra literature from the late 20th century and using it as a springboard for their modern, forward looking take on the classical questions in symbiosis. They center the story around metabolic interaction and make a good case with their data for showing metabolic interdependence, as evidenced by reduced nitrogen assimilation pathways in the symbiont, localization of symbiosis genes in the host, and light experiments showing that expression is tied to symbiont photosynthesis.

While we largely support the metabolic interaction storyline, we feel that the authors undersell some of their interesting findings – they seem to sweep them under the carpet. They show that some of the most differentially expressed genes in the host are in innate immune pathways and they even find that expression of the genes is not directly linked to symbiont productivity. Yet, the authors chose not to discuss innate immunity and its role in symbiosis at all, nor do they discuss the expression data – except to use them as almost a control for the metabolic gene expression patterns. Hence we feel that this is a missed opportunity and is a little surprising given Bosch and colleagues' expertise in the role of innate immunity in symbiosis. Similarly, the authors don't linger on the differences in expression when hosts are colonized by an inappropriate symbiont. We view this as another missed opportunity.

Another major component missing in this manuscript is a bird's eye view comparison to other relevant large-scale sequencing based studies. As authors pointed out in the Discussion, there are examples of endosymbioses that are based on nitrogen and carbohydrate nutrients, including those, such as coral-*Symbiodinium, Paramecium-Chlorella*, and salamander-green alga, that have been investigated by transcriptome approach. I'd thus encourage authors to discuss as to how similar and dissimilar the major molecular mechanisms of these distinct, yet ecologically comparable associations are.

Essential revisions:

Please expand the Discussion to include comparisons between your data and the much larger field of anthozoan-dinoflagellate symbiosis and other large scale genome analyses that are relevant.

*Reviewer #1:*

This manuscript describes genomic and transcriptomic evidence for the metabolic interdependence of the partners in the *Hydra-Chlorella* symbiosis. The work was carefully designed and executed and it is presented in a well-constructed and well-written manuscript.

The study casts the work in an evolutionary and historical light. The researchers do a good job of capturing the excellent original green *Hydra* literature from the late 20th century and using it as a springboard for their modern, forward looking take on the classical questions in symbiosis. They center the story around metabolic interaction and make a good case with their data for showing metabolic interdependence, as evidenced by reduced nitrogen assimilation pathways in the symbiont, localization of symbiosis genes in the host, and light experiments showing that expression is tied to symbiont photosynthesis.

While I largely support the metabolic interaction storyline, I feel that the authors undersell some of their interesting findings – they seem to sweep them under the carpet. They show that some of the most differentially expressed genes in the host are in innate immune pathways and they even find that expression of the genes is not directly linked to symbiont productivity. Yet, the authors chose not to discuss innate immunity and its role in symbiosis at all, nor do they discuss the expression data – except to use them as almost a control for the metabolic gene expression patterns. I feel that this is a missed opportunity and is a little surprising given Bosch and colleagues' expertise in the role of innate immunity in symbiosis. Similarly, the authors don't linger on the differences in expression when hosts are colonized by an inappropriate symbiont. I view this as another missed opportunity.

The authors have opted to keep the Discussion short and have not expanded it to include comparisons between their data and the much larger field of anthozoan-dinoflagellate symbiosis. There are many interesting parallels and differences that they could discuss – the most interesting to me being the differences in symbiont genome size the similarities for evidence loss of some pathways. Again, this is surprising given Satoh and Shinzato's expertise in coral genomics. I view this as another missed opportunity that undersells the impact of this study.

I found the naming of the organisms and the partnerships inconsistent and somewhat confusing. For example, the naming is different in Figure 1 and Table 1. Sometimes the symbiont strain name is used to represent its presence in the host – like in Table 1, but sometimes not, like in Figure 1. The authors also need to check their terminology throughout the manuscript and strive for consistency. For example, in the subsection “Symbiont-dependent *Hydra* genes are upregulated by photosynthetic activity of *Chlorella* A99”, the term 'aposymbiont' is confusing – I think here 'aposymbiotic' would be better. Sometimes the authors use photobiont, but others times just symbiont – etc. The term 'zooxanthellae' is no longer used in the literature. I would suggest changing to '*Symbiodinium*'.

Where is the complete dataset of differentially expressed genes summarized in Figure 1C? Will this list be made available to the reader?

There is a new paper by Sproles et al. MPE 2018 that extensively examines transporters in cnidarian-*Symbiodinium* symbioses and includes discussion of nitrogen transporters. This paper should be included in their Discussion.

In Figure 7, how would Gln be transported across the host symbiosome? There is no transporter depicted there.

N-numbers for the QPCR data should be included in the figure legend.

In the title, 'evolutionary' should be changed to 'evolutionarily'.

*Reviewer #2:*

This is an interesting study that describes *Hydra*-green algal symbiosis based on the use of large-scale genetic data. Their genome based investigation confirmed that the consortium is driven by nutrient exchanges: some animal genes identified in this study have implication in carbohydrate metabolism whereas the algal genome suggests its reduced competency in assimilation of inorganic source of nitrogen and an increased capacity to transport amino acids. The work produced, among other things, a high-quality genome assembly for the symbiotic *Chlorella* sp.

Abstract ("in an unbiased approach.."): While I agree with the authors that their microarray-based experiment with the host animal is an unbiased approach to investigate gene-level changes associated with symbiosis, it is certainly not the case for their study with the algal partner. For the alga, the authors focused specifically to select nitrogen metabolism related genes based on earlier studies. Thus, it is highly likely that other molecular aspects of algal responses to symbiosis may have been overlooked. Perhaps the authors could do an "unguided" comparison of the algal genome to its related, non-symbiotic genomes. If this proves to be too daunting, I suggest minimally the authors clearly indicate in the text the limitation of their experimental approach.

While reviewing this article, I came across the paper by Ishikawa et al. (2016)-which is referenced in this article. In the 2016 study, authors generated and compared transcriptomes of two different species of Hydra that were grown with or without green algal symbiont. In that RNA-seq based experiment, 1,890 contigs were found to be up-regulated and 2,261 to be down-regulated for symbiotic *H. viridissima* compared to the aposymbiotic animal. In contrast, Hamada et al. (this study) found much reduced sets of genes that are differentially regulated (423 and 256, respectively). As the two studies overlap quite a bit, it seems important that authors compare their results to the 2016 study data and address any major discrepancies that are to be found. For instance, are the DE genes identified in this study a subset of those found by Ishikawa?

Related to the above point, another major component missing in this manuscript is a bird's eye view comparison to other relevant large-scale sequencing based studies. As authors pointed out in the Discussion, there are examples of endosymbioses that are based on nitrogen and carbohydrate nutrients, including those, such as coral-*Symbiodinium, Paramecium-Chlorella*, and salamander-green alga, that have been investigated by transcriptome approach. I'd thus encourage authors to discuss as to how similar and dissimilar the major molecular mechanisms of these distinct, yet ecologically comparable associations are.

---

## [Author Response]

[…] While we largely support the metabolic interaction storyline, we feel that the authors undersell some of their interesting findings – they seem to sweep them under the carpet. They show that some of the most differentially expressed genes in the host are in innate immune pathways and they even find that expression of the genes is not directly linked to symbiont productivity. Yet, the authors chose not to discuss innate immunity and its role in symbiosis at all, nor do they discuss the expression data – except to use them as almost a control for the metabolic gene expression patterns. Hence we feel that this is a missed opportunity and is a little surprising given Bosch and colleagues' expertise in the role of innate immunity in symbiosis. Similarly, the authors don't linger on the differences in expression when hosts are colonized by an inappropriate symbiont. We view this as another missed opportunity.Another major component missing in this manuscript is a bird's eye view comparison to other relevant large-scale sequencing based studies. As authors pointed out in the Discussion, there are examples of endosymbioses that are based on nitrogen and carbohydrate nutrients, including those, such as coral-Symbiodinium, Paramecium-Chlorella, and salamander-green alga, that have been investigated by transcriptome approach. I'd thus encourage authors to discuss as to how similar and dissimilar the major molecular mechanisms of these distinct, yet ecologically comparable associations are.

Thank you for your comments and the reviews of our manuscript “Metabolic co-dependence drives the evolutionarily ancient *Hydra-Chlorella* symbiosis”. We the comments extremely helpful and constructive. In particular we thank you for your efforts in not letting us “sweep our interesting findings under the carpet”. Of course, innate immunity plays a major role. Therefore, in addition to the metabolic aspects of the symbiotic interactions described, we now added a chapter to present and discuss the differentially expressed immunity genes. We also followed all other suggestions in revising our work.

In order to provide a better overview of the *Hydra-Chlorella* symbiosis, we added new analyses of all differentially expressed *Hydra* genes and the gene loss in the native symbiont *Chlorella* A99. We also expanded the Discussion on generality and specificity of animal-algal symbiotic systems and its genome evolution and extended Figure 8 with a comparison between Hydra-Chlorella symbiosis and coral-*Symbiodinium* symbiosis.

We hope our responses provided below satisfactorily address all issues you have noted.

Essential revisions:Please expand the Discussion to include comparisons between your data and the much larger field of anthozoan-dinoflagellate symbiosis and other large scale genome analyses that are relevant.

Thank you for this excellent and constructive comment. In the previous version of our manuscript, we specifically focused on the metabolic genes, which are differentially expressed in native green *hydra* compared to both aposymbiotic *hydra* and *hydra* with non-native symbiotic alga NC64A. However, as the comment above suggested, there are still many other genes differentially expressed when comparing symbiotic and aposymbiotic state polyps. We now provide a list of all differentially expressed genes including their annotations in Supplementary file 1. Of these, we focused on the genes, which are known to be involved in the symbiosis of other Cnidarians (Table 1). We also added discussion about similarity and differences between the symbiosis of green hydra and other anthozoans (new Figure 8). In addition, we further analyzed the gene loss in *Chlorella* A99 and present the data in new Figure 5. We also added a comparison of the nitrogen metabolism pathway between *Hydra/Chlorella* and coral /*Symbiodinium* (Supplementary file 4) and expanded the discussion about symbiosis and genome reduction from the viewpoint of comparative genomics.

Reviewer #1:[…] While I largely support the metabolic interaction storyline, I feel that the authors undersell some of their interesting findings – they seem to sweep them under the carpet. They show that some of the most differentially expressed genes in the host are in innate immune pathways and they even find that expression of the genes is not directly linked to symbiont productivity. Yet, the authors chose not to discuss innate immunity and its role in symbiosis at all, nor do they discuss the expression data – except to use them as almost a control for the metabolic gene expression patterns. I feel that this is a missed opportunity and is a little surprising given Bosch and colleagues' expertise in the role of innate immunity in symbiosis.

As suggested by the referee, we have incorporated the data (list of best hit gene, GO and domain) of all differentially expressed genes between symbiotic and aposymbiotic *Hydra* (Supplementary file 1). Of these, we focused on the molecules, which are known to be involved in the symbiosis such as innate immunity, apoptosis and oxidative stress response, which are known to be involved in the symbiosis of other cnidarians such as coral and sea anemone (Table 1, subsection “Discovery of symbiosis-dependent *Hydra* genes”).

Similarly, the authors don't linger on the differences in expression when hosts are colonized by an inappropriate symbiont. I view this as another missed opportunity.

Thank you for this comment. We now extended our analyses by screening the genes differentially regulated in Hv_NC64A and present the data in Table 2. Strikingly, we did not find any genes related to immunity. Instead, the presence of the non-native symbiont NC64A seems to induce changes in some pathways of metabolite synthesis and ubiquitylation, which we briefly discuss in the Results section (subsection “Discovery of symbiosis-dependent *Hydra* genes”, last paragraph, Table 3, and Supplementary file 1). The question of their possible biological meaning is interesting itself but beyond the scope of our work, which primarily focusses on identification of genes specifically regulated by presence of the native symbiont *Chlorella* A99. Therefore, *Chlorella variabilis* NC64A served as a relevant control for symbiont specificity, since the alga is not naturally occurring as a symbiont of *Hydra*, but is principally able to establish a symbiotic relationship (with *Paramecium*).

The authors have opted to keep the Discussion short and have not expanded it to include comparisons between their data and the much larger field of anthozoan-dinoflagellate symbiosis. There are many interesting parallels and differences that they could discuss – the most interesting to me being the differences in symbiont genome size the similarities for evidence loss of some pathways. Again, this is surprising given Satoh and Shinzato's expertise in coral genomics. I view this as another missed opportunity that undersells the impact of this study.

Thank you for pointing this out. To address the question whether the gene loss in *Chlorella* A99 was species-specific for the green hydra symbiont or a general phenomenon in photosynthetic symbionts, we now searched the components of the nitrogen assimilation pathway in three genome-sequenced *Symbiodinium* species (subsection “The *Chlorella* A99 genome provides evidences for extensive nitrogenous amino acid import and an incomplete nitrate assimilation pathway”, fourth paragraph, Supplementary file 4). Further, we expanded the Discussion by adding a paragraph “More general lessons for animal-algal symbiosis” and a summary figure (new Figure 8B, C) to discuss common features and differences between the *Hydra-Chlorella* symbiosis and other symbiotic systems such as coral-*Symbiodinium*. We also added a more detailed discussion of genome reduction and degeneration of the nitrogen assimilation pathway in *Chlorella* A99 including comparison to other symbiotic and non-symbiotic organisms (subsection “Genome Evolution in symbiotic *Chlorella* sp. A99”).

I found the naming of the organisms and the partnerships inconsistent and somewhat confusing. For example, the naming is different in Figure 1 and Table 1. Sometimes the symbiont strain name is used to represent its presence in the host – like in Table 1, but sometimes not, like in Figure 1. The authors also need to check their terminology throughout the manuscript and strive for consistency. For example, in the subsection “Symbiont-dependent Hydra genes are upregulated by photosynthetic activity of Chlorella A99”, the term 'aposymbiont' is confusing – I think here 'aposymbiotic' would be better. Sometimes the authors use photobiont, but others times just symbiont – etc. The term 'zooxanthellae' is no longer used in the literature. I would suggest changing to 'Symbiodinium'.

We agree that the naming of organisms was confusing and corrected it to be consistent in the text, figures and tables. The abbreviations of green *hydra* and algae were unified to Hv_Sym, Hv_Apo and Hv_NC64A and the strain name respectively. We also unitized the various terms for symbiotic algae into “symbiont” and changed the term “zooxanthellae” to “*Symbiodinium*”.

Where is the complete dataset of differentially expressed genes summarized in Figure 1C? Will this list be made available to the reader?

We now added a complete list of all differentially expressed genes including annotations (best hit gene, GO and domain) as Table 1 and Supplementary file 1.

There is a new paper by Sproles et al. MPE 2018 that extensively examines transporters in cnidarian-Symbiodinium symbioses and includes discussion of nitrogen transporters. This paper should be included in their Discussion.

In the results about glucose transporter in the host and nitrate transporter in the symbiont, we referred to the paper by Sproles et al. (subsections “Discovery of symbiosis-dependent *Hydra* genes”, fourth paragraph; “Symbiont-dependent *Hydra* genes are expressed in endodermal epithelial cells and up-regulated by sugars”, last paragraph and “The *Chlorella* A99 genome provides evidences for extensive nitrogenous amino acid import and an incomplete nitrate assimilation pathway”, fourth paragraph) to compare the symbiosis of *Hydra* to that of anthozoans. We also added some other papers about *Symbiodinium* genome analysis (see aforementioned paragraph) as references.

In Figure 7, how would Gln be transported across the host symbiosome? There is no transporter depicted there.

Thank you for this comment. Gln is probably transported from the host cytoplasm into the symbiosome by an amino acid transporter. We now added a hypothetical amino acid transporter on the symbiosome in Figure 8A (previous Figure 7).

N-numbers for the QPCR data should be included in the figure legend.

We added the number of biological replicates as well as the number of animals used for each replicate in all figures displaying qRT-PCR data.

In the title, 'evolutionary' should be changed to 'evolutionarily'.

We agree and changed the title accordingly.

Reviewer #2:[…] Abstract ("in an unbiased approach.."): While I agree with the authors that their microarray-based experiment with the host animal is an unbiased approach to investigate gene-level changes associated with symbiosis, it is certainly not the case for their study with the algal partner. For the alga, the authors focused specifically to select nitrogen metabolism related genes based on earlier studies. Thus, it is highly likely that other molecular aspects of algal responses to symbiosis may have been overlooked. Perhaps the authors could do an "unguided" comparison of the algal genome to its related, non-symbiotic genomes. If this proves to be too daunting, I suggest minimally the authors clearly indicate in the text the limitation of their experimental approach.

Thank you for this constructive comment. We appreciate the advice and now performed an unguided genome analysis by comparison of all *Chlorella* A99 gene models to conserved genes among three free-living green algae. Thereby, we identified candidate genes, which are likely to be lost from the A99 genome (Supplementary file 3). Further, we examined the distribution of the candidate genes over different functional categories using Gene Ontology. The data are presented in new Figure 5 and suggest that many genes involved in “transport” were lost in the *Chlorella* A99 genome (Table 7).

While reviewing this article, I came across the paper by Ishikawa et al. (2016)-which is referenced in this article. In the 2016 study, authors generated and compared transcriptomes of two different species of Hydra that were grown with or without green algal symbiont. In that RNA-seq based experiment, 1,890 contigs were found to be up-regulated and 2,261 to be down-regulated for symbiotic H. viridissima compared to the aposymbiotic animal. In contrast, Hamada et al. (this study) found much reduced sets of genes that are differentially regulated (423 and 256, respectively). As the two studies overlap quite a bit, it seems important that authors compare their results to the 2016 study data and address any major discrepancies that are to be found. For instance, are the DE genes identified in this study a subset of those found by Ishikawa?

When comparing our analysis to that of Ishikawa et al., 2016, it is important to notice that different strains of green hydra were used. Further, also the platform to analyze the transcriptome (microarray in our study, and mRNA-seq in Ishikawa et al.), the threshold to identify differentially expressed genes (Fold change>2 and ttest pvalue<0.1 in our study, and false discovery rate (FDR) ≤ 0.1 in Ishikawa et al.), and the sampling conditions (three days starvation and six hours light exposure in our study, and seven days starvation in Ishikawa et al.) were different. However, we found some genes involved in oxidative stress response were differentially expressed in both our analysis and analysis by Ishikawa et al., and also transcriptome analysis in other animals. We refer to this in the Results section (subsection “Discovery of symbiosis-dependent *Hydra* genes”).

Related to the above point, another major component missing in this manuscript is a bird's eye view comparison to other relevant large-scale sequencing based studies. As authors pointed out in the Discussion, there are examples of endosymbioses that are based on nitrogen and carbohydrate nutrients, including those, such as coral-Symbiodinium, Paramecium-Chlorella, and salamander-green alga, that have been investigated by transcriptome approach. I'd thus encourage authors to discuss as to how similar and dissimilar the major molecular mechanisms of these distinct, yet ecologically comparable associations are.

As we described above (response to reviewer 1’s comment), we added a comparison to the other genome/transcriptome analyses, and a discussion on “More general lessons for animal-algal symbiosis” as well as a new summary figure (Figure 8B, C) to discuss the generality and specificity in animal-algal symbiotic systems. We also expanded the discussion on evolution of genome reduction by comparison of genome and ecology of other symbiotic and non-symbiotic organisms (subsection “Genome Evolution in symbiotic *Chlorella* sp. A99”).